# Targeting n-myristoyltransferases promotes a pan-*Mammarenavirus* inhibition through the degradation of the Z matrix protein

Xavier Carnec[1,2], Virginie Borges-Cardoso[1,2], Stéphanie Reynard[1,2], Heinrich Kowalski[3], Jean-Charles Gaillard[4], Mathieu Mateo[1,2], Jean Armengaud[4], Sylvain Baize[1,2]*

**1** Unité de Biologie des Infections Virales Emergentes, Institut Pasteur, Université Paris Cité, Lyon, France, **2** Centre International de Recherche en Infectiologie (CIRI), Université de Lyon, INSERM U1111, Ecole Normale Supérieure de Lyon, Lyon, France, **3** Center for Medical Biochemistry, Max F. Perutz Laboratories (MFPL), Medical University of Vienna, Vienna Biocenter (VBC), Vienna, Austria, **4** Laboratoire Innovations Technologiques pour la Détection et le Diagnostic (LI2D), Service de Pharmacologie et Immunoanalyse (SPI), Commissariat à l'Energie Atomique et aux Energies Alternatives, Bagnols sur Cèze, France

* sylvain.baize@pasteur.fr

**Data Availability Statement:** The mass spectrometry proteomics data have been deposited to the ProteomeXchange Consortium via the

## Abstract

Several Old World and New World *Mammarenavirus* are responsible for hemorrhagic fever in humans. These enveloped viruses have a bi-segmented ambisense RNA genome that encodes four proteins. All *Mammarenavirus* identified to date share a common dependency on myristoylation: the addition of the C14 myristic acid on the N-terminal G2 residue on two of their proteins. The myristoylation of the Z matrix protein is required for viral particle budding, while the myristoylation of the signal peptide to the envelope glycoproteins is important for the entry mechanism. Using Mopeia virus as a model, we characterized the interaction of the Z matrix protein with the N-Myristoyltransferases (NMT) 1 and 2, the two enzymes responsible for myristoylation in mammals. While both enzymes were capable to interact with Z, we showed that only NMT1 was important for the production of viral progeny, the endogenous expression of NMT2 being insufficient to make up for NMT1 in its absence. Using the high affinity inhibitors of NMTs, IMP1088 and DDD85646, we demonstrated a strong, dose dependent and specific inhibition at the nanomolar range for all *Mammarenavirus* tested, including the highly pathogenic Lassa, Machupo, Junin and Lujo viruses. Mechanistically, IMP1088 and DDD85646 blocked the interaction between Z and both NMTs, preventing myristoylation and further viral particle formation, egress and spread. Unexpectedly, we found that the matrix protein devoid of myristate, despite being fully translated, did not accumulate as the other viral proteins in infected cells but was instead degraded in a proteasome- and autophagy-independent manner. These molecules represent a new broad-spectrum class of inhibitors against *Mammarenavirus*.

## Author summary

Membrane anchoring of proteins is facilitated by protein acylation. Myristoylation is the covalent addition of a saturated C14 acyl to the N-terminal G residue of proteins by

PRIDE partner repository (https://www.ebi.ac.uk/pride/) with the dataset identifier PXD053439 and 10.6019/PXD053439.

**Funding:** This research was funded by an internal grant from Institut Pasteur and by the Laboratoire d'excellence Integrative Biology of Emerging Infectious Diseases (grant number ANR-10-LABX-62-IBEID) to SB. The funders had no role in study design, data collection and analysis, decision to publish, or preparation of the manuscript.

**Competing interests:** SB and XC are designated as inventors in pending European patent application No. 24160395 filed by the Institut Pasteur. The patent application covers the aspects described in the manuscript.

NMTs, a cellular feature exploited by many families of viruses to target their proteins to cellular membranes. *Mammarenavirus* have two myristoylated proteins out the four proteins they encode, the Z matrix protein and the stable peptide signal to the envelope glycoproteins GP1/2. We characterized the interaction of the Z matrix protein with NMT1 and NMT2 and found that during infection NMT1 has a predominant role for all *Mammarenavirus* tested. Using the recently developed NMT inhibitors IMP1088 and DDD85646, we provide strong evidences for a pan-*Mammarenavirus* inhibition of viral particle production within the nanomolar range. Mechanistically, the inhibition of the interaction between NMTs and Z blocked myristoylation that consequently promoted the post-translational degradation of the viral protein in a proteasome and autophagy independent manner.

## Introduction

The members of the *Mammarenavirus* genus family have a geographic, phylogenetic and serologic repartition with Old World and New World *Mammarenavirus* infecting tens of thousands of humans each year. These zoonotic viruses are primarily transmitted to humans in close contact with their rodent reservoirs. Several members of this family of viruses are responsible for severe hemorrhagic fever with a significant lethality rate such as Lassa virus (LASV) in Western Africa, Junin (JUNV) or Machupo (MACV) viruses in South America and represent a threat to public health. Moreover, survivors frequently experience sequelae-like deafness associated with LASV infection [1–3]. Epidemics of *Mammarenavirus* are difficult to identify because of the non-specific symptoms associated early in infection that facilitate the human to human transmission (often in a nosocomial environment), as exemplified with the unique Lujo virus (LUJV) epidemic reported to date [4]. Other *Mammarenavirus* have not been associated with human clinical cases, such as Mopeia virus (MOPV) in Southeastern Africa. Considerable efforts have been made to develop vaccines using different approaches with promising results, however with the exception of JUNV, no licensed vaccines are currently used in the field [5–7]. Meanwhile, the development of treatments based on novel drugs or drug repositioning provide new leads to treat infection [8–17].

*Mammarenavirus* have a genome consisting of two single stranded RNA molecules with an ambisense coding strategy. The L segment encodes for the RNA dependent RNA polymerase (Lpol) and the Z matrix protein. The S segment encodes for the precursor (GPC) of the envelope glycoproteins GP1-GP2 and their stable signal peptide (SSP) and for the nucleoprotein (NP). *Mammarenavirus* particles vary in sizes and shape. Their lipidic envelope, derived from the plasma membrane, has SSP-GP1-GP2 forming trimers at the surface while Z proteins line the inside of the particle. The viral genomic segments are encapsidated by NP and are associated with Lpol to form ribonucleoprotein complexes (RNPs). An interesting and peculiar characteristic of all *Mammarenavirus* is the presence of a G2 residue at the N-terminal of Z and SSP that rendered them both eligible to myristoylation, a covalent post-translational addition of the saturated C14 myristic acid.

The addition of myristate to proteins is catalyzed by Glycylpeptide N-tetradecanoyl transferases, or N-Myristoyltransferases (NMTs), to the N-terminal of a glycine G2 residue after methionine M1 removal by methionine aminopeptidases or by release of a N-terminal G residue after protein cleavage. NMTs then transfer myristate from the carboxylic moiety of a docked myristate-coA, to the free $NH_2$- of the glycine residue (G2) but also possibly to the free $NH_2$- of lysine side chain residues (K3) [18–20]. The process for G2 (and K3) residue

myristoylation occurs co- or post-translationally for a wide array of cellular proteins in eukaryotes from plants, fungi, protozoas to mammals [21]. As the number of identified myristoylated protein increases, the role of this lipidation commensurate in many biological and pathological processes [18,21–25]. In mammals, two isoenzymes with redundant enzymatic activity exist, NMT1 and NMT2, with both common and specific protein targets [19,20,24–26].

Viral proteins are known to undergo myristoylation at G2 residues, such as the Pr55[gag] polyprotein precursor and the Nef proteins for HIV-1 [27,28], the VP0 polyprotein of most picornaviruses [29–31] in humans, or the Geminivirus C4 protein in plants [32]. The myristoylation of Z and SSP for *Mammarenavirus* has an important impact for their respective functions. The absence of myristate on the Z matrix protein severely inhibits its inner membrane surface localization and self-capacity to bud from the plasma membrane [33,34]. The absence of myristate for SSP impairs, at the least, the efficacy of the viral or endosomal membrane fusion during the entry mechanism [35–37].

Several studies demonstrated the potency of structure guided designed compounds targeting the NMTs such as DDD85646 for treating *Trypanosoma brucei in vivo* [38], or inhibiting *Plasmodium falciparum* [39], and *Leishmania donovani* multiplication *in vitro* [40]. Another newly designed molecule, IMP1088, presents a nanomolar range inhibitory effect on both human NMTs and no off target effects compared to previously developed NMT inhibitors [41]. These molecules display a strong inhibitory effect *in vitro* against picornaviruses [42,43] or vaccinia virus infectivity [44]. Albeit different in structure, they mimic the N-terminal of the substrate peptide and occupy the binding site of the NMTs with a subnanomolar $K_D$ for IMP1088 [42,45].

We report here that both human NMTs are capable to interact with the Z matrix protein of *Mammarenavirus* but NMT1 is the main enzyme required for the production of infectious viral particles from infected cells. We also found that the IMP1088 and DD85646 compounds strongly inhibited the production of infectious viral particles in a dose dependent manner for all tested *Mammarenavirus*, including the highly pathogenic LASV, LUJV, MACV and JUNV. These compounds inhibited the interaction between Z and NMTs, thereby preventing myristoylation and the release of viral progeny. We also found that despite the absence of myristoylation the Z matrix protein is fully translated but degraded in a proteasome and autophagy independent manner.

## Results

### N-myristoyltranferase 1 and 2 interact with the Z matrix protein of *Mammarenavirus*

Human NMT1 and NMT2 catalyze the myristoylation of G2 and K3 residues of proteins. In order to visualize the Z matrix protein myristoylation during infection, we first setup an assay using an azido-dodecanoic acid (AzC12) metabolic analogue of the myristic acid for the biorthogonal ligation of the azido group with a terminal alkyne fluorescent reporter (Alk-AF647) through Cu[I]- catalyzed azide-alkyne cycloaddition ("click" reaction). To do so, A549 cells were infected at MOI 0.01 with three different recombinant MOPV: a WT (rMOPV WT), a N-terminal mCherryFlag expressing virus (rMOPV mCherryFlag, for which the NP ORF is modified as a mCherryFlag-P2A-NP ORF) or a virus expressing a Z protein carrying a C-terminal Flag (rMOPV Z-Flag). Of note, the rMOPV mCherryFlag is slightly attenuated compared to rMOPV WT and rMOPV Z-Flag viruses as previously described [46]. The AzC12 was added 24 h post infection (20 µM) and after 48 h of infection, cells were lysed to collect Whole Cell Extracts (WCE) and Flag tagged proteins were immunoprecipitated (Flag IP). Samples were then subjected to "click" reaction to label the protein-incorporated AzC12 with the Alk-

AF647. Proteins were then analyzed in SDS-PAGE conditions for the direct in gel detection of fluorescence ("in gel fluorescence"). The results of in gel fluorescence in Fig 1A showed that WCE contained equal amounts of AzC12 carrying proteins. In the Flag IP corresponding samples, only the incorporation of AzC12 in the Z matrix protein was present. We then tested in identical condition of infection whether NMT1 and/or NMT2 were able to interact with the Z matrix protein during infection. Fig 1B showed the presence of NMT1, NMT2, mCherry and Z protein in WCE analyzed by Western Blot. The same analysis of Flag IP extracts showed the co-immunoprecipitation of endogenous NMT1 or NMT2 in the Z-Flag condition but not in the WT or mCherryFlag control conditions.

To characterize further the interaction of the Z matrix protein with NMT1 and NMT2, HEK293T cells were cotransfected with plasmids encoding for the WT or a five alanine mutant (G2-K3-T4-Q5-S6/A) of the N-terminal of the Z-Flag MOPV and HA tagged mCherry, NMT1, NMT2 or ITCH, the latter a known interactor of Z [47,48] (Fig 1C). Western Blot analysis of Flag IP eluates confirmed the presence of an interaction between NMT1 or NMT2 with the WT Z-Flag MOPV protein while both enzymes did not interact with the Ala Z-Flag protein. The Z/ITCH interaction positive control was left unchanged regardless to the nature of the N-terminal of the Z protein. We next analyzed the consequences of a mutated catalytic site of NMT1 (N246L) and NMT2 (N248L) for the interaction with the WT Z-Flag MOPV protein (Fig 1D). The WCE from plasmids co-transfected HEK293T cells showed comparable expressions for WT and mutants NMT's. In Flag IP eluates, we observed a decrease of NMT1 N246L and NMT2 N248L mutants interacting with the Z-Flag protein compared to their WT counterparts. We also analyzed the capacity of the Z matrix proteins of LASV and JUNV to interact with overexpressed NMT1 and NMT2 in similar experiments (S1A Fig). Western Blot results on Flag IP eluates showed the presence of an interaction with the ITCH positive control as well as NMT1 and NMT2 for both Z-Flag proteins. Taken together, these results demonstrated that the Z matrix protein of *Mammarenavirus* interacted with NMT1 and NMT2 and this interaction relied on the N-terminal sequence of Z and residues of the catalytic site of NMT1 and NMT2.

Recent findings suggested that a lysine at position 3 can undergo myristoylation by NMT1 and NMT2, a residue present in MOPV but not in other Z matrix proteins of *Mammarenavirus* (S1B Fig) [19,20]. To verify whether the Z matrix protein of MOPV was myristoylated at K3, G2A and G2A/K3A mutations were introduced in the plasmid encoding for the WT Z-Flag. Along with control plasmids, HEK293T cells were transfected with these constructs and treated with AzC12. WCE were subjected to the cycloaddition of Alk-AF647 by click chemistry for in gel fluorescence and WB analysis. The results of S1C Fig showed that all Z constructs were expressed at similar level, but only the WT protein incorporated AzC12. Albeit theoretically possible, the Z matrix protein of MOPV was not K3 myristoylated.

## N-myristoyltransferase 1 is required for the production of infectious *Mammarenavirus*

Because other families of viruses have been shown to make a different use of NMT1 and NMT2, we evaluated the involvement of these enzymes in the biology of *Mammarenavirus*. To answer that question, we used HAP1 WT and HAP1 knock-out cells for either NMT1 (NMT1$_{ko}$) or NMT2 (NMT2$_{ko}$) [43]. After confirmation of the absence of respective proteins by western blot (Fig 2A), cells were infected at MOI 0.01 with MOPV, LASV, LCMV, LUJV, MACV or JUNV for 48 h and the cell culture supernatants of WT, NMT1$_{ko}$ or NMT2$_{ko}$ HAP1 cells were titrated. The results in Fig 2B showed that the production of infectious progeny from HAP1 NMT1$_{ko}$ was significantly decreased for all tested *Mammarenavirus* compared to

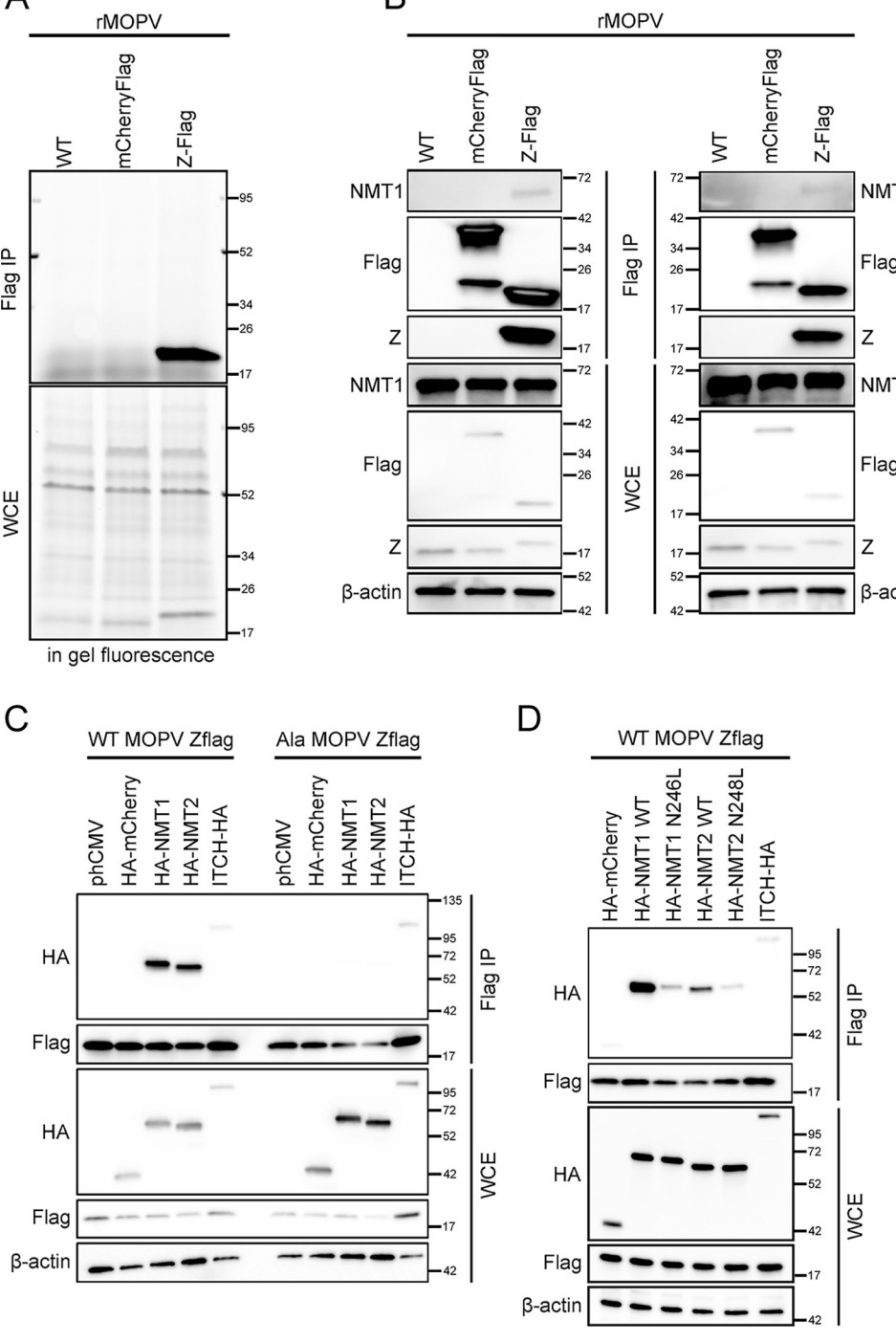

**Fig 1. *Mammarenavirus* Z protein interacts with NMTases.** Evidence of Z matrix protein myristoylation and interaction with NMT1 and NMT2 during infection (A-B). A549 were infected with rMOPV WT, mCherryFlag or Z-Flag at MOI 0.01 for 48 h, WCE were collected and Flag immunoprecipitated. (A) In gel fluorescence for the detection by biorthogonal ligation (click reaction) of Alk-AF647 with the AzC12 analogue of myristoylated proteins in WCE (bottom) or Flag immunoprecipitated proteins (top). (B) WCE and IP products from identically set experiments were analyzed by Western Blotting for the presence of NMT1, NMT2, Z, Flag and β-actin. (C-D) Characterization of the Z/NMT interaction. HEK293T cells were cotransfected for 24 h with the indicated plasmids before cell lysis and Flag immunoprecipitation. WCE and IP products were analyzed by Western Blot for the presence of Flag (WT or Ala mutant of the Z protein of MOPV), HA (NMT1, NMT2 and ITCH) and β-actin. Results shown are representative of two (A-B) and three (C-D) independent experiments.

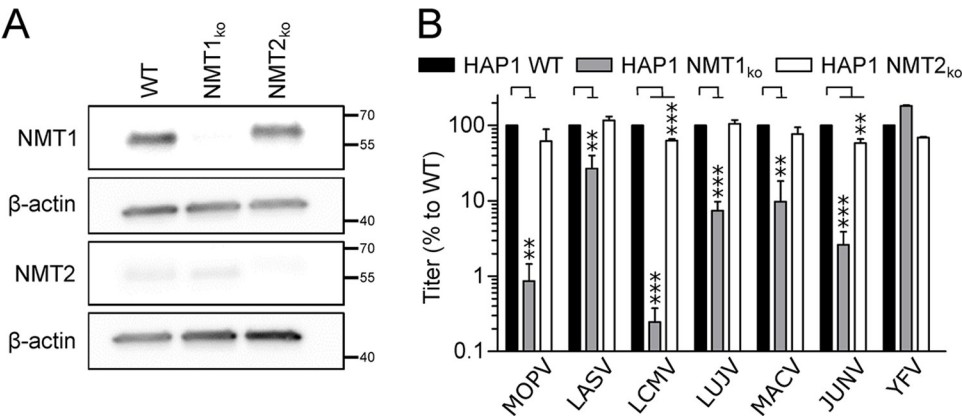

**Fig 2. NMT1 over NMT2 is a host dependency factor for *Mammarenavirus*.** (A) Western Blot analysis of WCE of HAP1 WT, NMT1$_{ko}$ or NMT2$_{ko}$ cells for the expression of NMT1, NMT2 and β-actin. (B) HAP1 WT, NMT1$_{ko}$ or NMT2$_{ko}$ cells were infected with MOPV, LASV, LCMV, LUJV, MACV, JUNV and YFV at MOI 0.01 for 48 h. Cell culture supernatants were titrated and the results of three independent experiments are represented as mean percentage of infectious titers +/- SEM of the HAP1 WTcells. $P$ <0.05 (*), <0.01 (**) and <0.001 (***).

the HAP1 WT cells: MOPV, LCMV, LUJV, MACV and JUNV presented a decrease reaching 90% to 99%, while LASV production was inhibited by 75%. In contrast, the absence of NMT2 had no effect for the multiplication of LASV and LUJV while the production of MOPV, LCMV and JUNV was inhibited by 30% to 40%. In order to verify whether the absence of NMT1 or NMT2 had an impact on viral multiplication regardless to the requirement of myristoylation for viral proteins, all three HAP1 cell lines were infected with the YFV 17D viral strain that has no known myristoylated proteins and found that its multiplication was not affected by the absence of NMT1 and decreased by 32% in the absence of NMT2 (Fig 2B). Our results showed that likewise other families of viruses, *Mammarenavirus* depended on NMT1 rather than NMT2 for multiplication.

## The paucity of NMT2 drives NMT1 dependency

NMT1 dependency for multiplication appeared as a common feature for several families of viruses [43,49]. To better explain this feature, we first quantified at 0 h, 24 h and 48 h the mRNA levels of NMT1 and NMT2 in A549 cells infected or not with rMOPV Z-Flag (MOI 0.01). The results in Fig 3A showed that mRNAs encoding for NMT1 were three to four times more abundant than mRNAs encoding for NMT2 and this ratio was conserved regardless to time and the presence or the absence of infection. In the matching WB, the expression of NMT1 or NMT2 seemed constant across time and unaffected by infection. Because double knock out of NMT1 and NMT2 is lethal for cells, we used HAP1 NMT1$_{ko}$ cells to investigate whether the complementation of the NMT1$_{ko}$ activity by the expression of NMT1 or NMT2 could restore the production of MOPV viral particles. HAP1 NMT1$_{ko}$ were transfected with control plasmids (empty vector or expressing HA-mCherry) or plasmids expressing N-terminal HA tagged NMT1 or NMT2 before infection with MOPV (MOI 0.01). WCE of transfected cells at time of infection were analyzed by western blot to verify protein's expression and cell culture supernatants collected 48 h post infection were titrated. The results in Fig 3B showed similar levels of expression of HA-NMT1 and HA-NMT2. The expression of the HA-NMT1 or HA-NMT2 increased MOPV viral titers 30 and 22 times respectively compared to the control plasmid. We studied the expression of NMT2 in the WCE from Fig 3B and found that i) the expression of HA-NMT1 allows the detection of a higher molecular weight version of

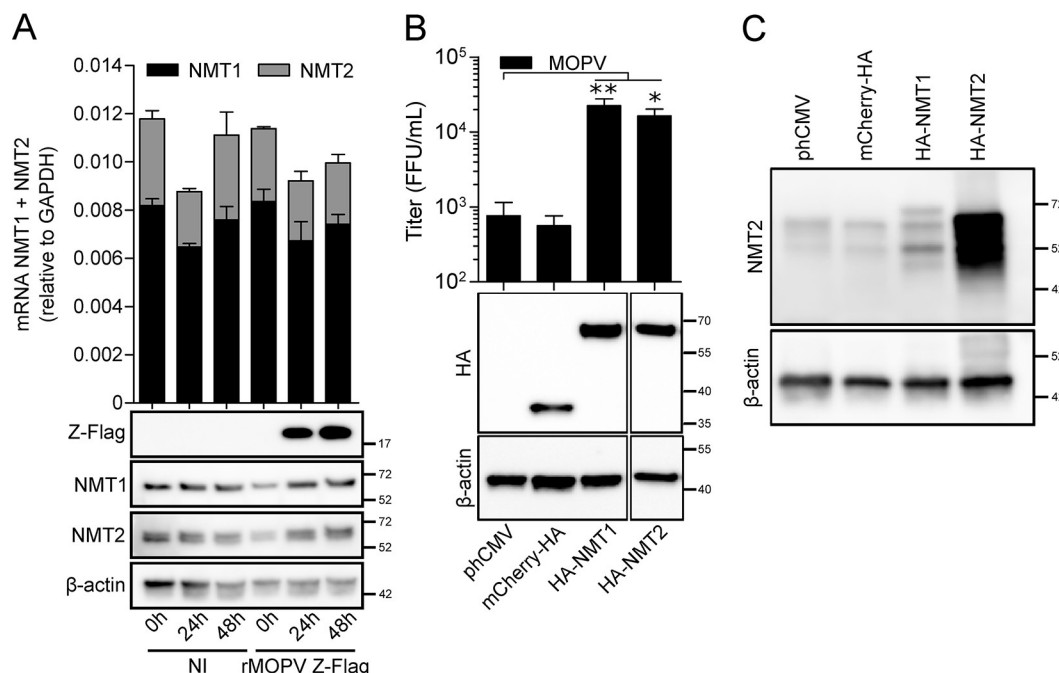

**Fig 3. Paucity of NMT2 drives NMT1 dependency.** (A) A549 cells were infected or not with rMOPV Z-Flag at MOI 0.01 and total RNA and WCE were collected at 0 h, 24 h and 48 h post infection. mRNA levels for NMT1 and NMT2 were quantified and standardized to GAPDH levels. WCE were analyzed by Western Blot for Z-Flag, NMT1, NMT2 and β-actin. Results are mean +/- SEM of three independent experiments. (B) HAP1 NMT1$_{ko}$ were transfected with the indicated control, HA-NMT1 or HA-NMT2 encoding plasmids for 24 h and then infected with MOPV at MOI 0.01 for 48 h. WCE collected at infection time were analyzed by Western Blot for the expression of HA tagged proteins and β-actin. HA-NMT1/2 lanes were brought closer from distant lanes. Cell culture supernatants were titrated and the results of four independent experiments are represented as mean +/- SEM and expressed as FFU/mL. (C) Endogenous NMT2 detection by Western Blot from the WCE of (B). $P < 0.05$ (*) and $< 0.01$ (**).

NMT2, possibly corresponding to low cross reactivity with NMT1 (Fig 3C, third lane) and ii) the expression of NMT2 is indeed highly increased (Fig 3C, fourth lane). Taken together, these results highlight that NMT2 mRNA and protein are rather less present in cells than NMT1's, but when its expression is increased, NMT2 is as capable as NMT1 to promote the recovery of viral production.

## The NMT1/2 inhibitors IMP1088 and DDD85646 are potent *in vitro* inhibitors of *Mammarenavirus* multiplication

The IMP1088 and DDD85646 (also known as IMP366) molecules were proven highly effective to inhibiting Rhinovirus, Enteroviruses or Poxvirus proliferation [42,43], [44]. We first assayed the potency of IMP1088, DDD85646 and C75, a fatty acid synthase inhibitor (tenfold serial dilutions for concentrations ranging from 1 nM to 10 μM) in A549 cells infected with MOPV, LASV, MACV or YFV 17D (MOI 0.01) for inhibiting the production of infectious particles in a multi-cycle infection experiment. Drugs were present in the culture media before, during and after infection and the cell culture supernatants collected 48 h post infection were titrated. Fig 4A, 4B and 4C (left panels) showed a dose dependent inhibition of the production of infectious particles for respectively MOPV, LASV and MACV in the presence of IMP108 and DDD85646 compared to C75 treated conditions. YFV titers were left unchanged regardless to the compounds and their respective concentrations. IMP1088 presented IC50 of 0.04 nM for MOPV, 1.31 nM for MACV and 2.96 nM for LASV while DDD85646 had IC50 of 8.45 nM for

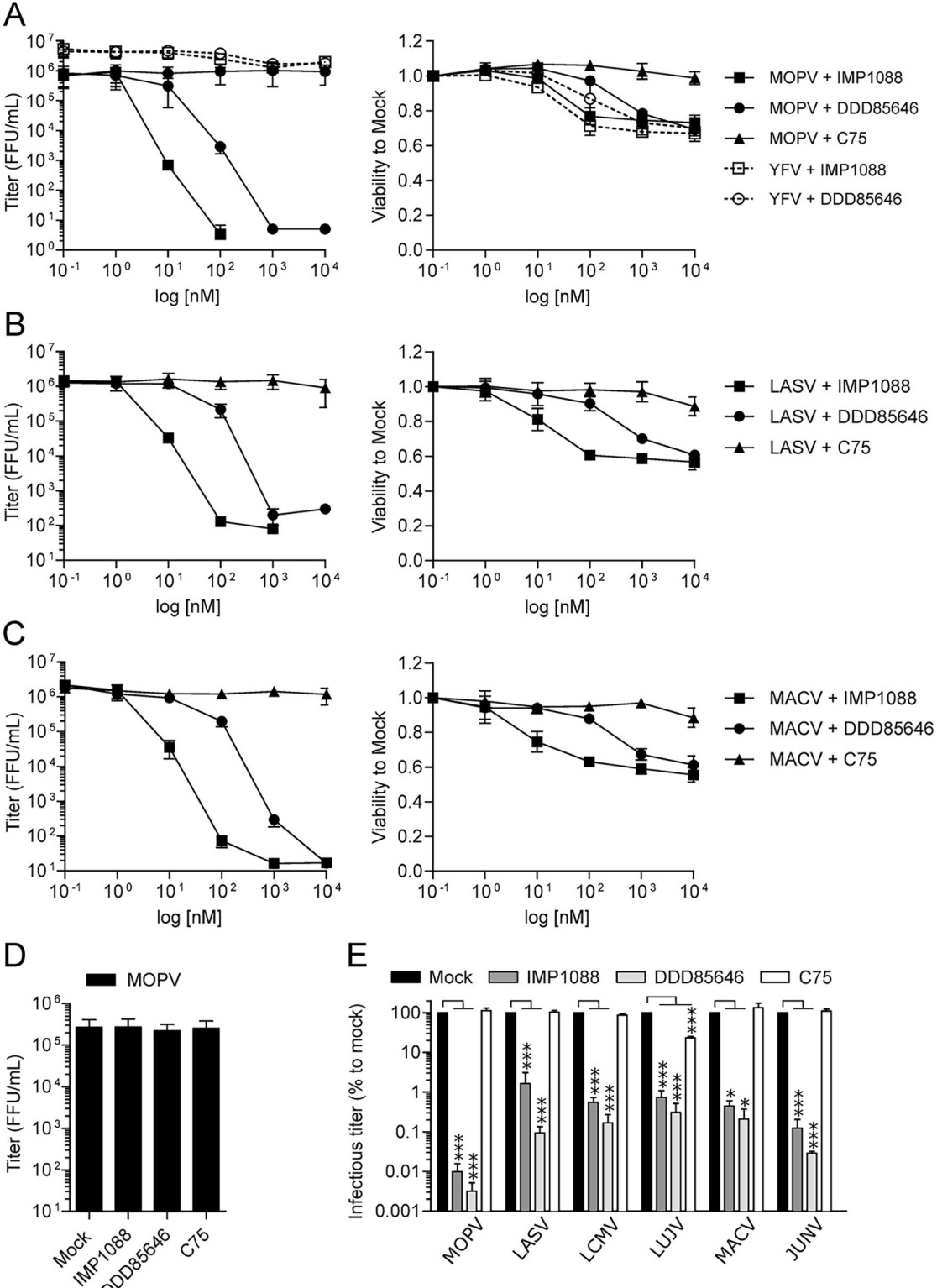

**Fig 4. IMP1088 and DDD85646 specifically inhibit *Mammarenavirus* multiplication.** (A-C) A549 cells were infected with YFV 17D, MOPV (A), LASV (B), or MACV (C) at MOI 0.01 for 1 h. Before, during and after infection, cells were treated or not with 10 fold dilutions of IMP1088, DDD85646 or C75 ranging from 1 nM to 10 μM. After 48 h of infection, cell culture supernatants were titrated (left panels) while cell viability was measured (right panels). The results of three independent experiments are represented as mean +/- SEM and expressed as FFU/mL for titrations and standardized to mock condition for viability. (D) Measure of the

potential virucidal effect of IMP1088, DDD85646 and C75 on MOPV particles. $10^5$ FFU of MOPV were incubated for 1 h with 10 μM of each compounds and were then titrated. The results of three independent experiments are represented as mean +/- SEM and expressed as FFU/mL. (E) Potency of IMP1088 and DDD85646 against MOPV, LASV, LCMV, LUJV, MACV or JUNV. A549 cells were infected (MOI 0.01) and treated or not with IMP1088 (20 nM), DDD85646 or C75 (500 nM) as described above. Cell culture supernatant were titrated and results of three independent experiments expressed as mean percentage +/- SEM to mock treated condition. $P < 0.05$ (*) and $< 0.001$ (***).

MACV, 9.84 nM for MOPV and 38.51 nM for LASV (S2A–S2D Fig). Due to the absence of inhibitory effect (IC50 > 3 μM for the three viruses), the C75 was used as a control drug throughout the study. The cell viability at the time of supernatant collection was measured through the reduction of Resazurin in Resofurin by NADPH for all concentrations of drugs and compared to their respective mock conditions (Fig 4A, 4B and 4C right panels). The viability of IMP1088 and DDD85646 treated cells was decreased at 100 nM and 1 μM respectively contrary to cells treated with C75 at the same concentrations. Therefore, the inhibition of the production of infectious virus occurred before reaching concentrations of IMP1088 and DDD85646 that impacted cellular viability.

To make sure that IMP1088, DD85646 or C75 had no virucidal activity, identical infectious units of MOPV were mixed with media containing 10 μM of each drug and incubated for 1 h à 37°C before titration. The results in Fig 4D showed no difference in infectious titers between mock and drug treated inocula. We next compared whether the drugs acted early or late during the viral cycle. A549 cells were treated or not with IMP1088 (20nM), DDD85646 or C75 (500nM) for 1 h before infection (Pre-infection), during infection with rMOPV WT at MOI 0.01 for 1 h (Infection only), after infection (48 h, Post-infection) or throughout the experiment (Complete). Cell culture supernatants were collected and titrated. The results of S2E Fig showed that an identical pattern of inhibition for the "Complete" and "Post-infection" conditions while the "Pre-infection" and "Infection only" conditions had minimum effects on the capacity of the virus to proliferate. Therefore, considering the biology of *Mammarenavirus* and the cellular targets of the inhibitors we concluded that IMP1088 and DDD85646 had no virucidal activity and inhibited late stages of the viral cycle and used the "Post-infection" treatment throughout the study.

To further evaluate the potency of both NMTs targeting drugs to inhibit the production of infectious *Mammarenavirus*, we infected A549 with MOPV, LASV, LCMV, LUJV, MACV or JUNV (MOI 0.01) and then treated the cells with IMP1088 (20 nM), DDD85646 and C75 (500 nM) and (Fig 4E). The production of all viruses was inhibited between 99% to 99,9% for both drugs excepted for LASV and IMP1088 which was only 97% inhibited. Collectively, our multi-cycle infection experiments at low MOI showed that IMP1088 and DDD85646 inhibited specifically and in a dose dependent manner within a nanomolar range the production of all tested *Mammarenavirus* in A549 cells without virucidal activity.

## IMP1088 inhibits Z interaction with NMT1 and NMT2 and its myristoylation

To characterize the molecular basis of the inhibition of *Mammarenavirus* multiplication in the presence of NMT inhibitors, we analyzed the possibility of IMP1088 to interfere with the interaction between Z and NMTs and inhibit its myristoylation (Fig 5). HEK293T cells were transfected with a MOPV Z-Flag encoding plasmid together with plasmids encoding WT HA-NMT1 (lanes 10–13), WT HA-NMT2 (lanes 6–9) or HA-ITCH (lanes 2–5), as well as empty plasmid and HA-mCherry controls in the presence (Fig 5A, lanes 1 and 14) or the absence (Fig 5B) of AzC12 (10μM). IMP1088 was then added to the cell culture medium at 10 nM, 100 nM or 1 μM final concentrations. After 24 h incubation, cells were lysed and WCE

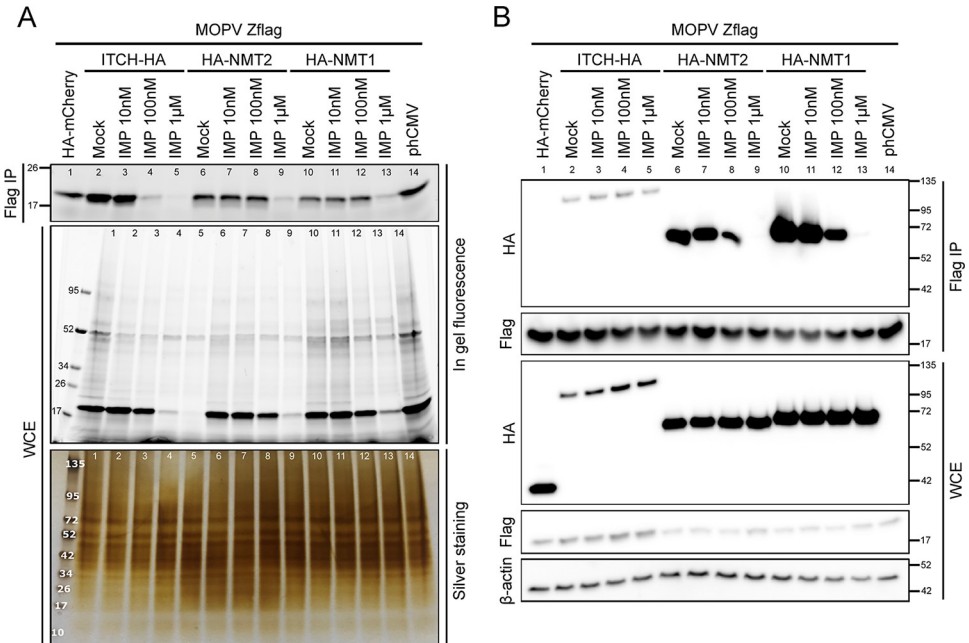

**Fig 5. IMP1088 inhibits myristoylation of Z through the inhibition of the interaction between the Z protein and NMTases.** HEK293T cells were transfected with a Z-Flag MOPV plasmid and HA-NMT1, HA-NMT2 or HA-ITCH plasmids (or control plasmids). Eight hours after transfection, cells were treated (A) or not (B) with AzC12 (10 μM) before mock or IMP1088 treatment (10 nM, 100 nM or 1 μM) for 24h. (A) WCE and IP eluates containing AzC12 incorporated proteins were subjected to "click reaction" with Alk-AF647 for in gel fluorescence detection of protein myristoylation and silver staining for total protein detection. (B) Non AzC12 treated WCE and IP eluates were analyzed by Western Blot as described in Fig 1. Results shown for A and B are representative of two independent experiments. Lane numbers [1–14] were added to match viewings between gels and panels.

subjected to Flag immunoprecipitation before "click" reaction to label the protein-incorporated AzC12 with the Alk-AF647. The in gel fluorescence analyses of Flag IP samples showed that the myristoylation of the Z-Flag matrix protein is inhibited in the presence of IMP1088 starting respectively at 100 nM (Fig 5A, lane 4) in the absence and 1 μM in the presence of ectopic expression of HA-NMT2 or HA-NMT1 (Fig 5A, lanes 9 and 13). The analyses of the corresponding WCE also showed a dose-dependent inhibition of the myristoylation of the Z matrix protein as well as of cellular myristoylated proteins (Fig 5A, WCE). The proteins present in the gel were then stained with silver. The results showed that the inhibition is restricted to myristoylated proteins because no inhibition of protein staining could be observed throughout the gel. Western Blot analysis on WCE and IP eluates from identically set experiments (but without Az-C12) showed that in the presence of IMP1088, the interaction between the MOPV Z matrix protein and NMT1 or NMT2 is inhibited from 100 nM in a dose dependent manner while the interaction of Z with ITCH was left unchanged. It is worth noticing that in these ectopic conditions of expression, a higher concentration of IMP1088 was required for the inhibition of Z myristoylation by HA-NMT1/2 compared to the HA-ITCH (Fig 5A, lanes 4, 8 and 12). Taken together, these results showed that IMP1088 inhibited the myristoylation of the Z matrix protein by blocking the interaction with NMT1 and NMT2.

## IMP1088 and DDD85646 inhibit viral propagation *in vitro*

To delineate the effect of IMP1088 and DDD85646 treatment in inhibiting *Mammarenavirus*, we carried out immunofluorescence assays (IFA) for the presence of the NP protein as a

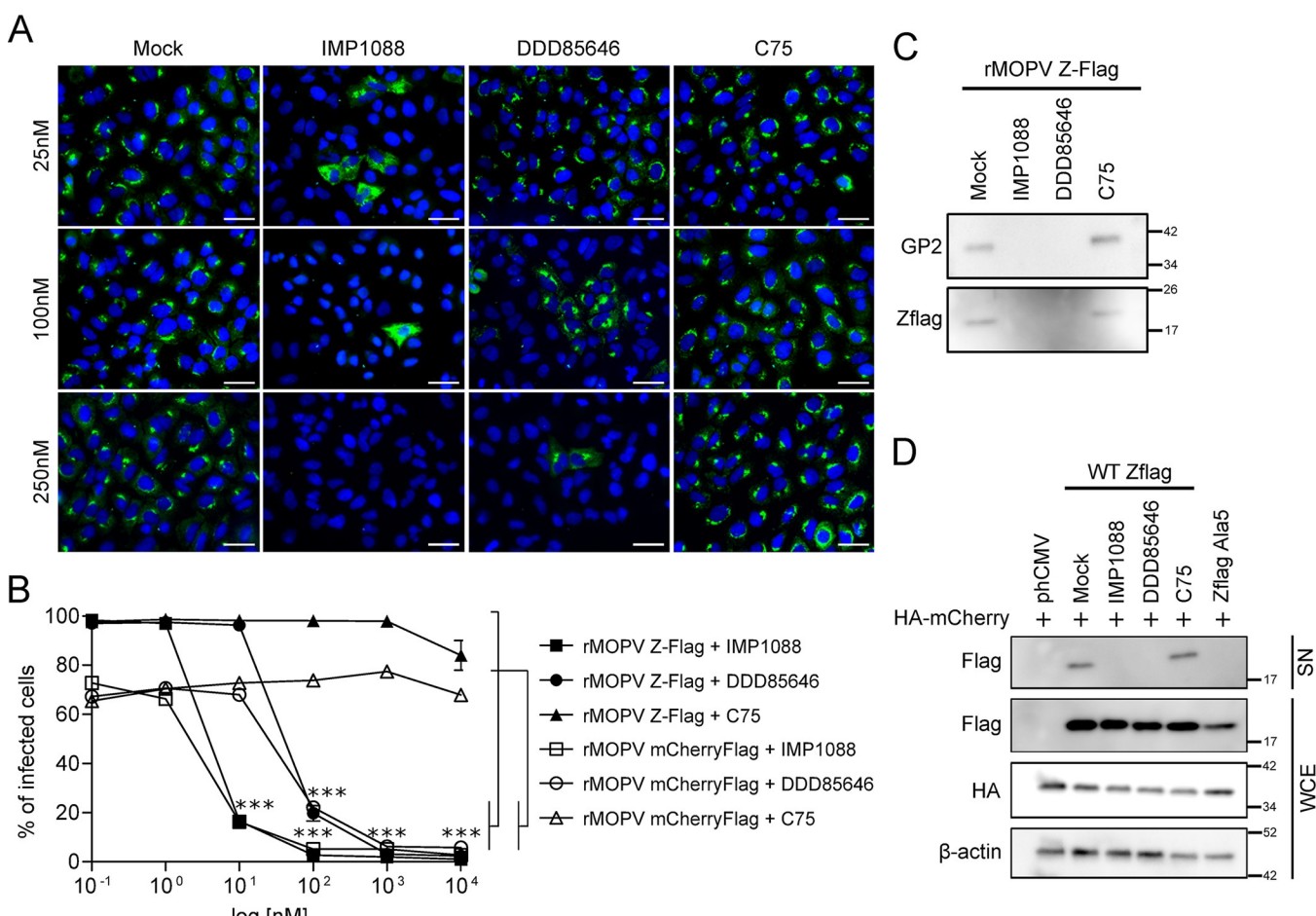

**Fig 6. IMP1088 and DDD85646 block cell to cell spread through the inhibition of viral particle egress.** (A) A549 cells were infected with MOPV at MOI 0.01 and mock treated or treated with IMP1088, DDD85646 or C75 at 25, 100 or 250 nM. After 48 h of infection, cells were fixed and stained to detect intracellular NP (green channel) and counterstained with DAPI. Results shown are representative of two independent experiments. Scale bar 20 μm. (B) A549 cells were infected with rMOPV Z-Flag or rMOPV mCherryFlag at MOI 0.01 and treated with 10 fold dilutions of IMP1088, DDD85646 or C75 as in Fig 4A. After 48 h of infection, cells were collected, fixed and stained with an anti-Flag-APC conjugated mAb to detect intracellular Z-Flag or mCherryFlag and analyzed by FACS. The results of three independent experiments are represented as mean +/- SEM and expressed as percentage of infected cells. (C) Supernatants from A549 infected with rMOPV Z-Flag (MOI 1) and treated with IMP1088 (50 nM), DDD85646 or C75 (500 nM) were ultracentrifuged (sucrose cushion 20% w/v). Pellets were analyzed by Western Blot for the presence of GP2 or Z-Flag proteins. (D) HEK293T cells were transfected with indicated plasmids and treated or not with IMP1088, DDD85646 or C75 for 24 h. Cell culture supernatants were ultracentrifuged as in (C). WCE and pellets analyzed by Western Blot for the presence of Z-Flag, mCherryFlag and β-actin. Results in C-D are representative of two independent experiments. $P < 0.001$ (***).

marker of infection in A549 infected with MOPV at MOI 0.01 for 48 h in the presence of different concentrations of IMP1088, DDD85646 or C75 (25, 100 and 250 nM). The results in Fig 6A showed a decreased in the number of NP positive cells treated with NMT's inhibitors compared to mock or C75 treated cells. To confirm these results, A549 were treated with the same compounds (tenfold dilutions ranging from 1 nM to10 μM) and infected with either the rMOPV Z-Flag or the rMOPV mCherryFlag. After 48 h of infection, we quantified the percentage of infected cells through intracellular Flag staining by flow cytometry (FACS). As shown in Fig 6B, the percentage of Flag positive cells with the C75 treatment remained stable for both viruses and comparable to the mock condition except for a slight decrease at 10 μM. The percentage of Flag positive cells infected with rMOPV Z-Flag or rMOPV mCherryFlag decreased in a dose dependent manner with IMP1088 and DD85646 treatment, reaching 12–

15% at 10 and 100 nM respectively and 1–2% at higher concentrations, a proportion of cells that might reflect the number of infected cells expected by the inocula. The decrease of NP positive cells in IFA and Z-Flag and mCherryFlag positive cells in FACS experiments indicated that both drugs affected the dissemination of viral progeny from initially infected cells. We also verified whether the IMP1088 treatment was involved in the IFN response. A549 cells were infected or not with rMOPV Z-Flag (MOI 1) and treated or not with IMP1088 (50 nM) and total RNAs were extracted at 0 h, 24 h and 48 h post infection for β-IFN mRNA quantification (S3). β-IFN mRNA transcription was not induced by IMP1088 treatment itself and infection triggered only a modest β-IFN mRNA production beginning 24 h post infection regardless to the presence or the absence of IMP1088. At 48 h post infection, the β-IFN mRNA transcription was highly increased (>100 fold) in the presence of IMP1088 compared to the mock treated condition. Taken together, these results suggested that the absence of viral propagation induced by the treatment with IMP1088 promoted an β-IFN response.

## Compounds targeting NMTs inhibit *Mammarenavirus* particle egress

Because two viral proteins of *Mammarenavirus* are myristoylated, we used reverse genetics to clarify their respective importance for MOPV by introducing G2A mutations in the Z and/or GPC ORFs. The results of the S4A Fig showed that the G2A in the Z ORF did not allow the rescue of rMOPV contrary to the G2A in the GPC ORF. The growth kinetics of rMPOV WT and rMOPV GPC G2A in Vero E6 cells showed similar results, indicating that the absence of myristate in the SSP was not determinant for this virus (S4B Fig). We then analyzed the viral particles released from infected cells that were treated or not with IMP1088 (50 nM), DDD85646 or C75 (500 nM). The cell culture supernatants from A549 cells infected with rMOPV Z-Flag at MOI 1 for 48 h were submitted to ultracentrifugation and the pellets were analyzed for the presence of the Z and GP2 proteins. The results in Fig 6C showed that Z and GP2 were present in supernatants from mock or C75 treated cells, while supernatants from IMP1088 and DDD85646 treated cells were devoid of both proteins. To confirm these results, we tested the effect of NMT's inhibitors on the self-budding capacity of the Z matrix protein (Fig 6D). HEK293T cells were transfected with plasmids expressing the WT MOPV Z-Flag or the N-terminal Ala mutant (Z-Flag$_{A5}$), as well as a plasmid expressing the HA-mCherry as positive control for transfection/expression. After transfection, cells were mock treated or treated with IMP1088 (50 nM), DDD85646 or C75 (500 nM) for 24 h before supernatants collect for ultracentrifugation followed by Western Blot analysis. In WCE, we found that regardless to the presence of drugs, HA-mCherry and MOPV Z-Flag proteins were expressed at similar levels. We noted a decreased steady state expression of the MOPV Z-Flag$_{A5}$ mutant. Ultracentrifuged cell culture supernatants contained WT Z-Flag only in mock or C75 conditions whereas no Z protein could be detected in the presence of IMP1088 or DDD85646 as well as Z-Flag$_{A5}$ mutant. Altogether, these results highlighted that IMP1088 and DDD85646 blocked the production of viral particles from infected cells because of the inhibition of the Z matrix protein budding capacity.

## IMP1088 and DDD85646 promote Z matrix protein disappearance from infected cells

The results obtained in Fig 6B showed a decrease in the percentage of infected cells in NMT's inhibitor treated cells. During these experiments, we noticed that the mean fluorescence intensity (MFI) of infected cells decreased for Z-Flag but remained unchanged for mCherryFlag. Because these findings relied on 1% of infected cells, we asked whether these results could be reproduced with higher percentages of infected cells and reached biological and statistical

significance. A549 cells were infected with rMOPV Z-Flag at MOI 0.01 for 48 h before mock or IMP1088 treatment (100 nM) for 48 h, a higher concentration of IMP1088 being necessary in these experiments to circumvent adsorption of the drug on the coverslips. Cells were then stained for the intracellular presence of GP2, NP or Z proteins for IFA (Fig 7A). In mock condition, we found comparable amounts of cells expressing GP2, NP or Z proteins. In the presence of IMP1088, we did not observe a decrease in the number of cells expressing GP2 or NP compared to mock condition but the number of cells expressing the Z matrix protein was drastically decreased.

In order to confirm our observations, we infected A549 cells with rMOPV Z-Flag or rMOPV NP-HA (N-terminal HA tagged NP ORF in the S segment) at MOI 0.01 for 48 h before treatment with drugs as set in Fig 6C. It is worth mentioning that the rMOPV NP-HA virus is slightly attenuated. Using flow cytometry for intracellular detection of Z-Flag or NP-HA, we quantified the percentage of infected cells (Fig 7B) and the MFI (Fig 7C) of these cells at 0 h, 24 h and 48 h post treatment. Our results showed that at the starting point of drug treatment, 70% of the cells infected with rMOPV Z-Flag were Flag positive and 30% of the cells infected with rMOPV NP-HA were HA positive. After 48 h of treatment, mock and C75 treated cells were nearly 100% Flag positive and HA positive cells were 55 to 60%, while IMP1088 and DDD85646 treated cells were 80% Flag positive and 40% HA positive. In Fig 7C, we found that the MFI of HA positive cells remained unchanged throughout the experiment regardless of the treatment used. Conversely, we observed a significant decrease of the Flag MFI in IMP1088 and DDD85646 treated cells after 24 h and 48 h of treatment compared to mock and C75 treated cells. These results were confirmed by Western Blot analysis on WCE from identically set experiments (Fig 7D). We observed a constant and similar increase presence of NP-HA in cells for all conditions. In mock and C75 treated cells, we also observed an increase of Z-Flag presence but in IMP1088 and DDD85646 treated cells, the presence of the Z-Flag protein decreased constantly during treatment. To confirm these results, we analyzed by tandem mass spectrometry the WCE of A549 infected for 48 h with MOPV (MOI 1) treated or not with IMP1088 (50 nM). The results in Fig 7E showed no difference in the detection of NP and GP1/2 derived peptides while the presence of Z derived peptides decreased by almost a tenfold in cells treated with IMP1088 compared to the mock condition. It is worth mentioning that we did not observe variations for NMT1 derived peptides and never detected NMT2 derived peptides in both conditions. We readily verified whether this decrease in the Z protein levels in infected cells originates from a decrease in Z RNA. To do so, total RNAs were extracted from the rMOPV Z-Flag samples of Fig 7A–7C experiments and RNA for NP and Z were quantified and standardized to GAPDH levels (Fig 7F). The results showed a general and non-significant twofold decrease of NP and Z RNA at 24 h and no differences in the levels of NP RNA or Z RNA at 48 h post treatment. RNA encoding for the Z matrix protein are present in infected cells treated with NMT inhibitors to similar extend with levels measured in control conditions. Taken together, our results showed that the presence of NMT's inhibitors promoted a post transcriptional and specific decrease expression for the Z matrix protein while the other viral proteins continued to be present in infected cells.

## NMTs inhibition promotes the degradation of the Z matrix protein

The decrease of the Z matrix protein abundance in NMT inhibitor treated cells might be linked to a co-translational and/or a post translational regulation step. To evaluate whether the mRNA encoding for the Z matrix protein was still translated during NMT inhibitor treatment, we generated the rMOPV Z-Flag-P2A-mCherryHA virus, a recombinant MOPV for which the Z-Flag ORF is modified by the 3' addition of a P2A sequence followed by a C-terminal HA

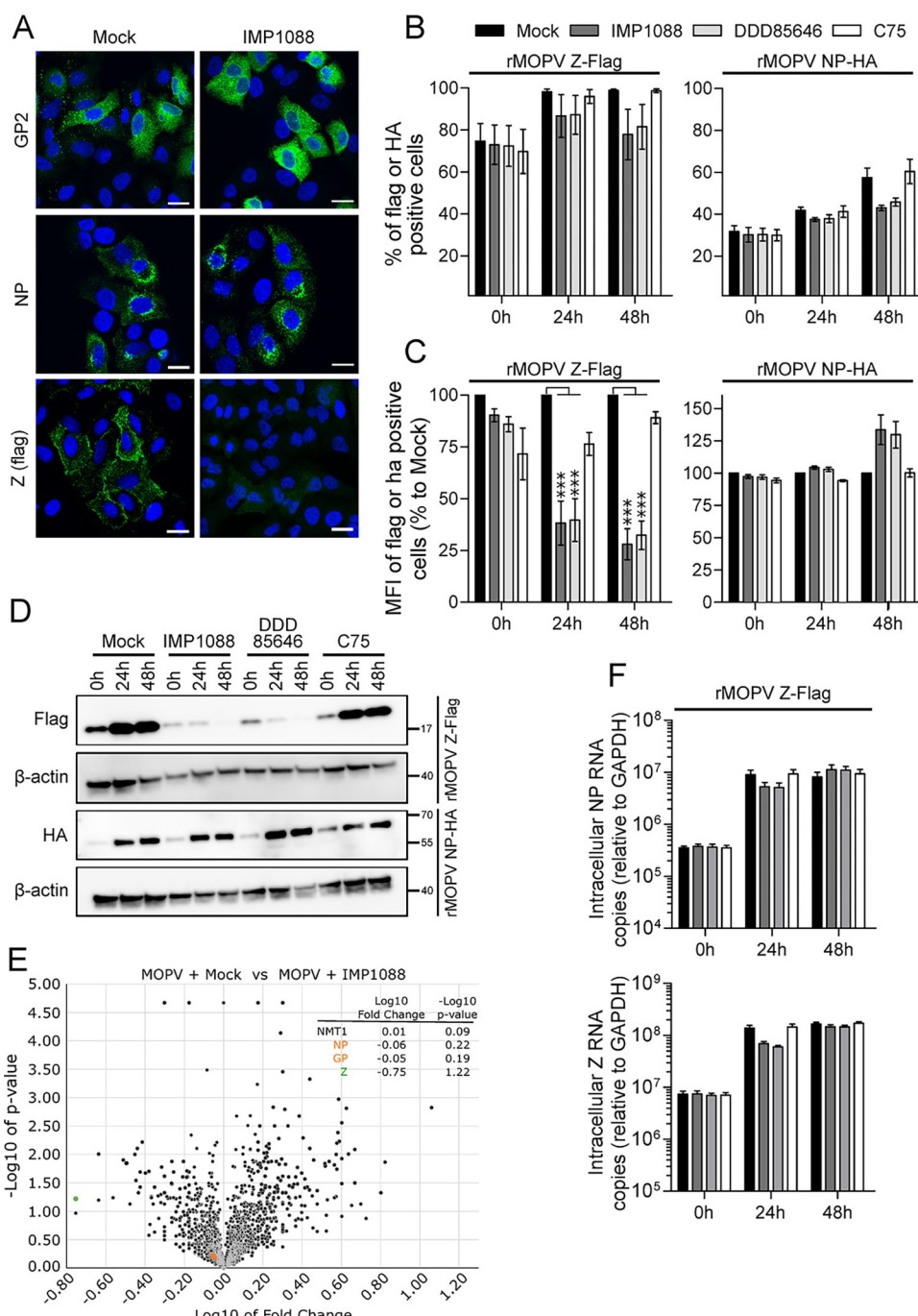

**Fig 7. IMP1088 and DDD85646 inhibit viral particle release and promote Z matrix protein disappearance from infected cells.** (A) A549 cells were infected with rMPOV Z-Flag at MOI 0.01 for 48 h before mock or IMP1088 treatment (50 nM) for the following 48 h. Cells were then stained for the expression of GP2, NP or Z proteins (green) and DAPI. Scale bar 10 µM. (B-C) A549 cells were infected at MOI 0.01 with rMPOV Z-Flag (left panel) or rMOPV NP-HA (right panel) for 48 h before mock or IMP1088 (50 nM), DDD85646 or C75 (500 nM) treatment for 48 h. Permeabilized cells were stained with anti-Flag APC or anti-HA FITC and analyzed by FACS. The results of five (rMPOV Z-Flag) and four (rMPOV NP-HA) independent experiments are expressed in (B) as the mean +/- SEM of the percentage of Fag or HA positive cells and in (C) as the mean of fluorescence intensity (MFI) +/- SEM of Flag or HA positive cells (percentages relative to the mock condition). (D) A549 cells were infected and treated as in (B-C) and WCE were analyzed by Western Blot for the presence of Z-Flag, NP-HA and β-actin. (E) Volcano plot representation of tandem mass spectrometry and comparative proteomics analyses of WCE from A549 cells infected with rMOPV Z-Flag at MOI 1 and treated or not with IMP1088 (50 nM) for 48 h (three independent experiments). (F)

Quantification of intracellular mRNA encoding for NP (top) or Z (bottom) from A549 cells infected with rMOPV Z-Flag at MOI 0.01 and mock or IMP1088, DDD85646, C75 treated. mRNAs were quantified at 0 h, 24 h and 48 h post treatment. Results are mean +/- SEM of four independent experiments expressed as mRNA copies standardized to GAPDH levels. *P* <0.001 (***).

tagged mCherry ORF (Fig 8A). Hence, from the same mRNA molecule expressed in an infectious context, it is possible to monitor the expression of both proteins during infection. A549 cells were infected with rMOPV Z-Flag-P2A-mCherry-HA at MOI 2 and cells were collected at different time points to evaluate Z-Flag and mCherryHA expression by flow cytometry. The percentage of infected cells was similar regardless to the cells treatment with drugs compared to mock, starting from 15 to 25% 5 h post infection and rapidly increasing to 80% 10 h post infection before reaching 95 to 99% from 14 h to 24 h post infection (Fig 8B). The MFI for Flag positive cells increased constantly in mock and C75 treated cells contrary to IMP1088 and DDD85646 treated cells for which we observed MFI decreases of 47% at 10 h, 73% at 14 h and 75% at 24 h post infection compared to control conditions (Fig 8C). The MFI for HA positive cells also increased in control conditions, but the decrease of MFI observed in IMP1088 and DDD85646 treated cells was 15% at 10 h, 25% at 14 h and 33% at 24 h post infection (Fig 8D). Western Blot analysis of WCE extracts from cells identically infected and treated recapitulated these results (Fig 8E). Taken together, our results indicated that the Z-Flag-P2A-mCherryHA mRNA is fully translated regardless of the treatment, but the two proteins had different fates once cleaved: compared to control conditions the mCherryHA abundance in NMTs inhibitors treated cells remains mostly unchanged contrary to the Z-Flag abundance. Therefore, in an infectious context when the myristoylation is inhibited, the Z matrix protein is fully translated before being degraded.

## Proteasome and autophagy are not involved in the degradation of the Z matrix protein upon treatment with IMP1088

An N-degron quality control mechanism of protein N-myristoylation has been identified through the recognition of non-myristoylated N-terminal sequences by the Cul2/5 Cullin-RING E3 ligases (CRL) and their substrate adaptors ZYG11B and ZER1 that target N-terminal glycine degrons to the proteasome for degradation [50]. We asked whether the treatment of infected cells with IMP1088 might trigger the CRL driven degradation of the Z matrix protein to the proteasome pathway with the use of the proteasome inhibitor MG132. We also used Chloroquine to inhibit the autophagy, another catabolic pathway for proteins. A549 cells infected with rMPOV Z-Flag virus at MOI 2 were treated or not with IMP1088 (50 nM) and treated with MG132, Chloroquine (1 μM or 10 μM) or with a mix of both drugs for 24 h. WCE were collected and analyzed by western blot for the presence of Z-Flag matrix protein and β-actin. We also detected HSP70 and LC3 I/II in order to confirm the efficacy of the drug treatments [51,52] (Fig 9). The results showed that treatment with MG132 or Chloroquine at 10 μM, increased HSP70 presence (lanes 5–6 and 13–14) and promoted lipidation of LC3-I into LC3-II (lanes 9–10 and 13–14) respectively. In the presence of IMP1088, the presence of the Z matrix protein could not be restored with either MG132, Chloroquine or a combination of both (even lane numbers 4 to 14). Taken together, this result indicated that the non-myristoylated Z matrix protein is degraded through an autophagy and proteasome-independent mechanism.

## Discussion

The myristoylation of proteins is catalyzed by two enzymes in mammals, NMT1 and NMT2. The addition of myristate to N-terminal glycine residues is either co-translational or acquired

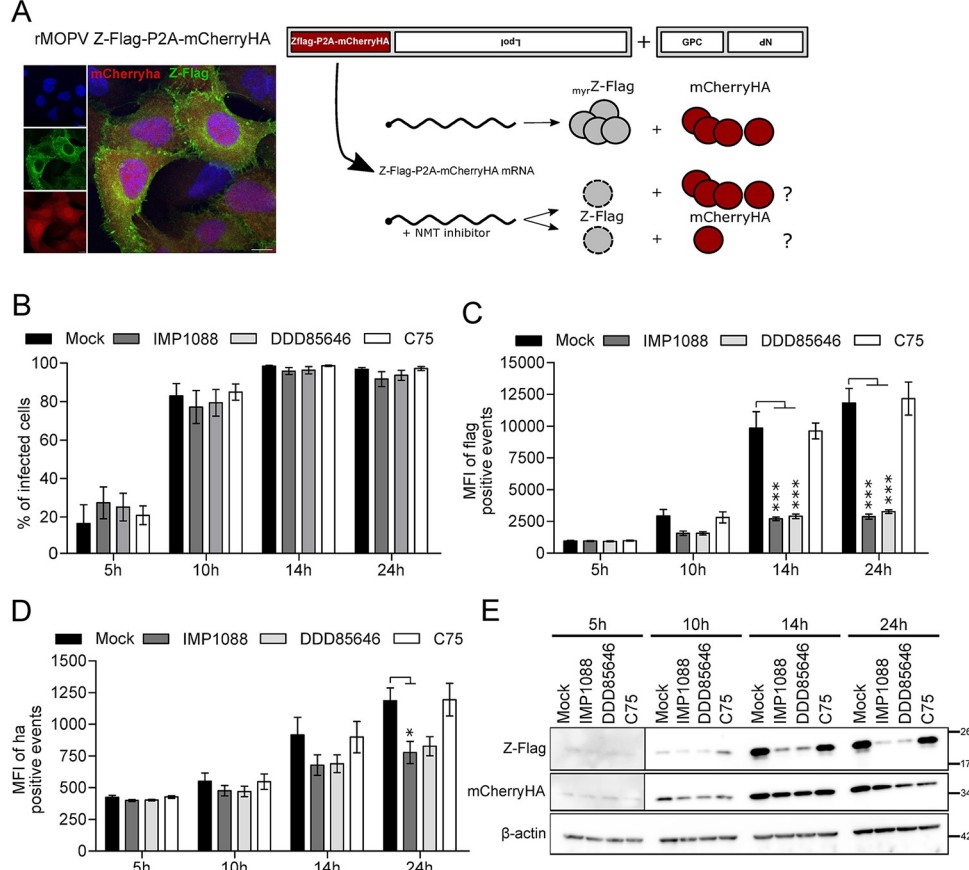

**Fig 8. The Z matrix protein is degraded in a post-translational manner.** (A) Reverse genetics strategy for the expression of Z-Flag and mCherryHA from the MOPV Z ORF and representative immunofluorescence of A549 cells infected with the rMOPV Zflag-P2A-mCherryHA depicting the expression of Z (green) and mCherry (red) and DAPI counterstaining (scale bar 5μm). (B-E) A549 cells were infected with the rMOPV Zflag-P2A-mCherryHA at MOI 2 and Mock treated or treated with IMP1088 (50 nM), DDD85646 or C75 (500 nM). Cells were collected at 5 h, 10 h, 14 h and 24 h post infection for intracellular staining with anti-Flag APC or anti-HA FITC and analyzed by FACS. The results of four independent experiments are in (B) the mean +/- SEM of the percentage of Fag positive cells and in (C) and (D) are the mean of fluorescence intensity (MFI) +/- SEM of Flag or HA positive cells respectively (percentages relative to the mock condition). (E) WCE of experiment set as in (B-D) analyzed by Western Blot for the presence of Z-Flag, mCherryHA and β-actin. Results shown are representative of three independent experiments. *P* <0.05 (*) and <0.001 (***).

upon release from protein cleavage, the latter chiefly related to apoptotic events. Many families of viruses make use of myristoylation for protein lipidation that facilitate the production of infectious progeny. As myristoylation is ubiquitous and mandatory for cell survival, viruses will always benefit from NMT activity, but they become highly dependent to a single post-translational modification reaction to fulfill viral production. Among these families, *Mammarenavirus* have two out of their four encoded proteins that require myristoylation.

NMT1 and NMT2 have redundant enzymatic activities [19,20] but clearly differentiate in term of biological importance and functions. NMT1 seems involved in more cellular processes than NMT2, but studies emphasizing the role of NMT2 are scarce [22–24,53,54]. We first showed that during infection the Z matrix protein interacted with NMT1 and NMT2, the enzymatic nature of this interaction rendering it likely difficult to grasp and detect [55]. We characterized further this interaction in an ectopic manner and found that it relied on the canonical features for myristoylated proteins, i.e. an N-terminal favorable context for Z in the

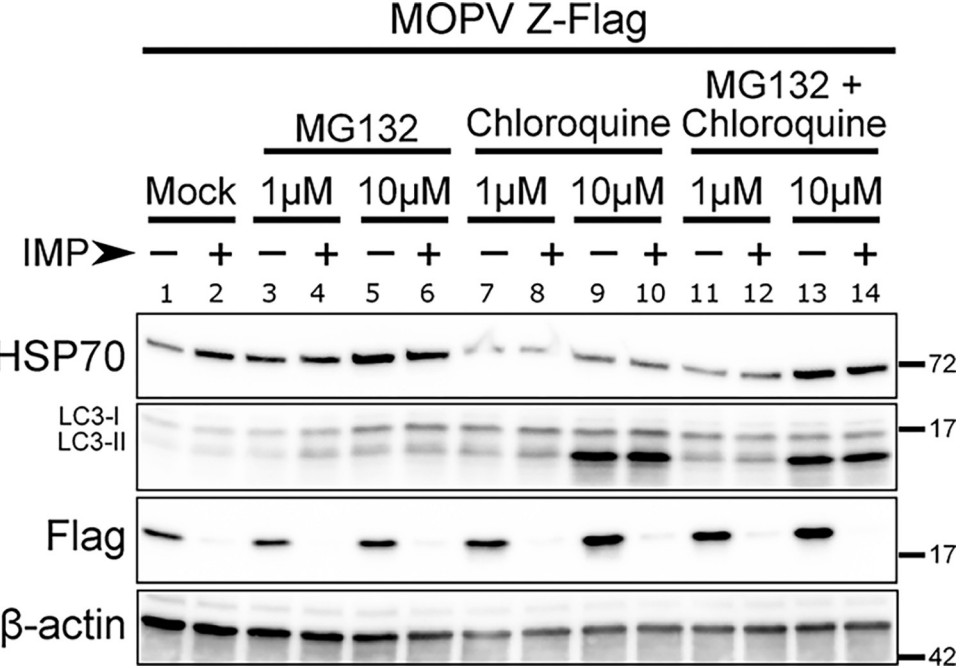

**Fig 9. The IMP1088 driven degradation of the Z matrix protein is independent of the proteasome and the autophagy catabolic pathways.** A549 cells were infected with rMOPV Z-Flag at MOI 2 and subsequently treated or not with IMP1088 (50 nM). MG132, Chloroquine (1 μM or 10 μM) or a mix of both drugs was readily added and incubated for 24 h. WCE were analyzed by Western Blot for the presence of HSP70, LC3I/II, Z-Flag and β-actin. Results are representative of two independent experiments.

active site of the enzymes. Both enzymes are present at the protein levels in our cellular extracts but we could not clearly differentiate the location of NMT1 and NMT2, nor evidence the location of their interaction with the Z matrix protein during infection. Paradoxically to these results but in accordance with results related to other families of viruses [43,49], we showed that NMT1 is the main enzyme required for the production of *Mammarenavirus* progeny *in vitro*. Therefore, to support viral production, three non-mutually exclusive hypotheses may explain the predominance of NMT1: i) endogenous expression of NMT2 is scarce, ii) NMT2 wrongly locates to myristoylate enough of the Z matrix protein and/or iii) NMT2 is regulated post-translationally.

Because endogenous NMT2 was unable to promote a robust viral production in the absence of NMT1 while being capable to interact and myristoylate Z, we reasoned that an increased expression of NMT2 might compensate the absence of NMT1. We experimentally confirmed the capacity of NMT2 to replace NMT1 to support viral production when plasmid driven protein expression is promoted. This result led us to conclude that the abundance of NMT1 over NMT2 is likely the determinant of its predominance but we could not clearly identify what are the molecular determinant(s) that support our result. *In situ* detection of endogenous NMT1 and NMT2 was hindered by the strong homology between the two enzymes. It is reported that they are able to interact and co-localize with each other in lung cancer cell lines [57]. Since Z is co-translationally myristoylated, it would likely be in close vicinity of both enzymes. Therefore, the location of NMT2 is unlikely the main explanation for the widespread use of NMT1 by viruses.

We measured mRNA levels of both enzymes in A549 cells and found that there was three to four times more mRNA encoding for NMT1 than NMT2 in cells regardless to the presence or

the absence of infection, a result similar to observations in other cell lines [25]. Studies showed that the abundance of mRNA encoding NMT1/2 was important for development. Heterozygotes NMT1$^{+/-}$ mice had monocytic lineage dysfunction and homozygotes NMT1$^{-/-}$ mouse embryos had only 5% of the total NMT activity compared to NMT1$^{+/+}$ ones [22,53]. The expression of NMTs is also post-transcriptionally regulated by miR-182 (and analogues) that might influence the abundance of NMT1 and NMT2: downregulation by expression of miR-182 and upregulation by its inhibition [56]. Our results, added to others, evidenced that mRNA levels explain, at least in part, the predominance of NMT1 over NMT2, but more studies are needed to investigate whether and how miRNA regulation would take part in the predominance of NMT1.

The regulation of the enzymatic activity of NMT1 and NMT2 by post-translational modifications may also explain the observed paramountcy. NMT1 undergoes *in vitro* tyrosine phosphorylation by Src tyrosine kinases Lyn and Fyn at residues 173 and/or 180 that are important for its enzymatic activity [57]. Such residues are also present in NMT2 but not studied. In a more recent study, wherein a recombinant NMT1 is produced in bacteria, Y180 is not phosphorylated yet the enzyme remained fully active even when the residue is modified to phenylalanine [19]. The Y180 localizes in the Ab loop of NMT1 and engages a hydrogen bond with the water molecule also bonded with the "to be myristoylated" G2 residue inside the catalytic site of the enzyme. Therefore, the addition of a phosphate group inside the active site of the enzyme seems rather dispensable for its activity and sterically difficult to reach. When overexpressed in HepG2 cells, the Akt/PKB is also capable to phosphorylate NMT1 at yet unidentified serine or threonine residue(s) that drastically decreased its activity [58]. Phosphorylation of NMT1 and/or NMT2 by Akt/PKB or other post-translational modifications by cellular enzymes could lead to a fine tuning of the global NMT activity and explain the differences between the two enzymes.

Inhibitors of myristoylation were long associated with low specificity and high cytotoxicity [34,41] but new high specificity and low cytotoxicity compounds are now available and effective to inhibit human pathogens [38–40,42,43], [44]. Here we provide the first evidence of *Mammarenavirus* inhibition using IMP1088 and DDD85646. These compounds inhibit in a dose dependent and specific manner at the nanomolar range all *Mammarenavirus* tested with nearly a tenfold stronger potency of IMP1088 over DDD85646, a molecule with already a strong efficacy. The addition of these compounds to cells during infection inhibited the production of *Mammarenavirus* progeny as cell culture supernatants did not contain viral particles. Mechanistically, we showed that these compounds blocked the interaction between the N-terminal of the Z matrix protein and the active site of NMTs. The absence of myristoylation of the Z matrix protein is known to abrogate its self-capacity to bud [33,34], but in an infectious context, we show for the first time a collateral effect regarding the absence of myristoylation through the decrease in the abundance of the Z matrix protein. This decrease did not originate from the transcripts encoding for Z, as the levels of viral RNA encoding for Z remained comparable during infection whether the cells were treated or not with the inhibitors. To clarify if the decrease of the Z matrix protein in NMT inhibitors treated cells occurred during or after translation, we engineered a virus that encoded for both Z and mCherry from the same mRNA molecule. Our results showed that the mCherry kept being produced in infected cells while the level of Z matrix protein simultaneously decreased in the presence of inhibitors. Since the P2A cleavage is incomplete, the Z-Flag-P2A-mCherryHA polypeptide would be fated for degradation as the myristoylation free Z-Flag, thereby explaining the minor decrease of the mCherry abundance in NMT inhibitor treated cells. Nonetheless, the observed decrease of Z abundance in infected cells occurred at a post translational step when myristoylation is inhibited. This N-terminal "myristate free" of Z constitutes an N-terminal degron

aimed for proteasome degradation through the Cullin-RING E3 ligases and their substrate adaptors [50]. Surprisingly, Z abundance was not restored when the proteasome and/or the autophagy were inhibited, a result reminiscent of the degradation of the L72A mutant of the Z matrix of LCMV [59]. A recent study has shown that the myristoylation of the Rpt2/PSMC1 targets a small fraction of the 26S proteasome to the plasma membrane and various organelles where it plays a key role for endomembrane homeostasis and protein trafficking. HIV Gag processing and p24 production were also affected by the absence of functional membrane associated proteasomes (MAPs) indicating that MAPs might be involved in viral protein trafficking and processing [60]. Collateral to the disappearance of the Z matrix protein in infected cells, the treatment with IMP1088 also promoted MAP inhibition [60] while MG132 inhibited the remaining proteasomal activity. Taken together, these results strongly indicated that the proteasome is unlikely responsible for the degradation of the Z matrix protein. In the absence of myristoylation, the "free" N-terminal G residue of the Z matrix protein became available to co-translational acetylation but it has been shown that this modification rarely occurred for G residues and also led to ubiquination and proteasome degradation [61]. Deciphering the cellular mechanism(s) responsible for the degradation of the Z matrix protein in the absence of myristoylation would require to investigate new leads from the field.

To date, the chemical inhibition of myristoylation was not reported to affect viruses devoid of myristoylated protein(s). It is interesting to notice that the consequences of NMT inhibition for myristoylation dependent viruses led to the inhibition of late stages and/or impairment of the early stages of their biology, the latter being largely a consequence of the former. When cells infected with Coxsackievirus B3 were treated with DDD85646, the absence of VP0 myristoylation led to a profound decrease of viral production while the viral particles that egressed from these cells were poorly infectious [43]. Vaccinia virus produced in IMP1088 treated cells were also poorly infectious due to the absence of myristate on the L1 protein that reduced membrane fusion and core release [44]. In the presence of myristoylation inhibitors, the infection by *Mammarenavirus* would also be impacted at early and late stages by the absence of myristate of SSP and Z respectively. Previous works have shown that the absence of myristoylation in the SSP of Candid-1 strain of JUNV has little or no effect on the capacity of the virus to assemble and bud from infected cells but is critical for the viral entry mechanism [36,37]. Our results showed that GPC G2A rMOPV had little or no attenuation compared to the WT rMOPV. The role of SSP myristoylation might therefore be different and not redundant between Old World and New World *Mammarenavirus*. The Z matrix protein, when myristoylated, orchestrates assembly and budding. Whether the absence of myristate on Z would permit some release of viral progeny, the absence of myristate on SSP would have different consequences depending on the targeted virus. Accordingly, this inhibition let few possibilities for escape mutants to emerge considering the central role of myristoylation in the biology of *Mammarenavirus* and the cellular substitutes for a lipid driven anchoring of the Z matrix protein in a comparative context (and also for the SSP with GP1 and GP2 during viral entry).

Aside from inhibiting viral particle budding and egress, the absence of Z may have multiple consequences for infected cells. Likely, and among others, the L polymerase would remain catalytically active and RNA synthesis will continue [62,63], [64,65] while the inhibition of RLR dependent interferon production would stop [66]. Interestingly, we found that NP and GP2 (and likely GP1) were still produced in infected cells treated with NMT inhibitors. Until clearance by immune cells, a pool of infected cells would remain stable and constitute small viral antigen factories for the antiviral response. Our results indirectly supported this hypothesis: once treated with IMP1088, we showed that the pool of primarily infected cells remained stable while the β-IFN mRNA transcription increased 48 h post infection. The timing of this

increased transcription likely originated from bystander cells in response to β-IFN from infected cells.

Another compound targeting NMTs, PCLX-001, has shown promising results in inhibiting tumor growth *in vitro* and promoting tumor regression in a mouse model of B-cell xenograft *in vivo* [67]. To date, the *in vivo* efficacy of NMTs targeting compounds has not been investigated for viruses. *In vitro*, the strong inhibitory effect of PLCX-001, IMP1088 and DDD85646 on NMT was fast, complete and occurred at concentrations that did not impact cellular viability. As NMT activity is mandatory for cell survival, further developments are needed to study tolerance, bioavailability and pharmacokinetics *in vivo*. Combining one NMT inhibitor with others compounds that have different cellular or viral targets might represent a new strategy for broader range antiviral therapies.

## Materials and methods

### Cells and viruses

BHK T7/9 cells were used to rescue recombinant viruses and were maintained as described before [68]. Vero E6 cells, A549 and HEK293T cells were grown in Glutamax Dulbecco Modified Eagle's Medium (Life Technologies) supplemented with 5% FCS, 1% Hepes, and 0.5% Penicillin-Streptomycin (P/S). HAP1 WT, $NMT1_{ko}$ or $NMT2_{ko}$ were maintained in Iscove Modified Dulbecco Medium, 10% FCS, 0.5% P/S (Life Technologies). The MOPV strain AN21366 was used for infection and to establish the rMOPV reverse genetics system. The Lassa AV, Lujo, lymphocytic choriomeningitis virus (LCMV) WE, MACV Carvallo and JUNV Espindola strains were used in this study as well as the Yellow Fever Virus 17D. All experiments with LASV, LUJV, MACV and JUNV were performed in the Jean Mérieux INSERM BSL-4 facility.

### Plasmid constructs

The four-plasmid strategy for the reverse genetics system of MOPV is described elsewhere [68]. The C-terminal Flag-tag sequence of the Z ORF in the L segment of MOPV and the N-terminal HA-tag of the NP ORF in the S segment were added using site directed mutagenesis. The mCherryFlag-P2A-NP ORF in lieu of NP ORF was generated by overlapping PCR and cloned into a ΔNP S segment of MOPV using BsmBI restriction sites. To generate the Z-Flag–P2A-mCherry-HA ORF in the L segment, the sequence encoding the P2A and cloning cassette (using a BsmBI based cloning strategy [68]) were introduced before the stop codon of the Z ORF by site directed mutagenesis. The mCherryHA sequence was then cloned in frame between the P2A sequence and the stop codon. The WT NMT1 (1491bp), the WT NMT2 (1497bp) ORFs were cloned in phCMV with an N-terminal HA tag. The Z ORFs of MOPV, LASV and JUNV were cloned in phCMV with a C-terminal Flag tag. All mutations in NMT1, NMT2, GPC and Z ORFs were introduced using the site-directed mutagenesis according to the manufacturer's instructions (Agilent). All plasmid constructs were verified by sequencing.

### Virus rescue and titration

The rescue procedure and viral amplification of recombinant viruses for MOPV were previously reported [68]. The second passage of MOI 0.01 infected Vero E6 cells constituted the viral stocks for all experiments. The absence of mycoplasma contamination during cell culture and viral stocks was confirmed by MycoAlert test (Lonza). For viral titration, cells were fixed with PBS 4% formaldehyde (FA) and permeabilized with PBS 0.1% Triton X-100. Immunostaining for LASV, LCMV, LUJV and the MOPV was carried out by a monoclonal antibody to

the NP protein and a phosphatase alkaline conjugated polyclonal goat anti-mouse antibody (Sigma). Immunostaining for the MACV and JUNV was carried out by polyclonal rabbit anti-Z matrix protein (AgroBio, France) and a phosphatase alkaline conjugated polyclonal goat anti-rabbit antibody (Sigma). 1-Step NBT/BCIP with suppressor substrate was used to stain viral focuses (Thermo Fisher scientific, Waltham, MA). Results for titration are expressed as Focus Forming Unit per mL (FFU/mL).

## Viability assay

The cell viability at the time of supernatant collection was assayed through the reduction of Resazurin in Resofurin by NADPH in cell culture by measuring light emission at 570nM.

## Chemical compounds and inhibitors

The IMP1088, 5-[3,4-difluoro-2-[2-(1,3,5-trimethyl-1H-pyrazol-4-yl)ethoxyphenyl]-N, N,1-trimethyl-1H-indazole-3-methanamine (CAS 2059148-82-0), DDD85646, 2,6-dichloro-4-[2-(1-piperazinyl)-4-pyridinyl]-N-(1,3,5-trimethyl-1H-pyrazol-4-yl)-benzenesulfonamide (CAS 1215010-55-1) and C75, tetrahydro-4-methylene-2R-octyl-5-oxo-3S-furancarboxylic acid (CAS 191282-48-1) were purchased from Cayman Chemicals. MG132 and Chloroquine were from Sigma. All drugs were resuspended in DMSO except Chloroquine resuspended in water, stored at -20°C and diluted in culture medium for inhibition assays.

## Quantitative RNA analysis

For RT-qPCR experiments, total RNAs were isolated from cells using the Rneasy Mini kit according to the manufacturer's instructions (Qiagen) and a supplementary DNase step using the Turbo DNA free kit Ambion was added following total RNA extraction (Thermo Fisher Scientific). Superscript III was used for cDNA synthesis and amplification was performed using the Gene Expression Master Mix kit (Applied Biosystems, Thermo Fisher Scientific). Taqman probes for NMT1, NMT2 and GAPDH were purchased from Applied Biosystems (Thermo Fisher Scientific). An in-house primer/probe mix was used for the detection β-IFN mRNA. For viral RNA quantification, RNA standards for NP and Z were set as described in [68]. Quantitative RT-qPCR was performed using SensiFAST Probe No-ROX One-Step kit (Bioline), using 5'-GTCAAGCGTTCTTTGGGAATG-3', 5'-TCCAGAAAGACATAGTTTG TAGAGG-3'primers and a FAM-TTCCTTTCCCCTGGCGTGTCA-BHQ1 probe for NP and 5'-CAGTAATACCAGATGCCAGGG-3', 5'-CTGAAGGGTAGTTTGTGTTTGC -3'primers and FAM-CACAGTGTCAGATAGGTGCCCCAT-BHQ1 probe for Z. The expression of all genes was standardized to that of the GAPDH gene and expressed as fold induction relative to GAPDH. Runs of qPCR assays were performed in a LightCycler 480 (Roche Diagnostics).

## Immunofluorescence Assay and Flow Cytometry

For immunofluorescence assays (IFA), coverslip seeded A549 were fixed in PBS 4% paraformaldehyde (PFA) before permeabilization with PBS 0.1% Triton X-100. Cells were then saturated with PBS 5% FCS and stained with monoclonal antibodies to Flag (clone M2, Sigma), NP (clone L52-54-6A, in house production) or GP2 (clone KL-AV-1G12, [69]) and an AF488 conjugated goat anti-mouse antibody (Invitrogen, Thermo Fisher Scientific). Coverslips were mounted with ProLong Gold Antifade containing DAPI for nuclei staining. Images were acquired using an Axiovert200M epifluorescent microscope (Fig 6) and a LSM 980 confocal microscope (Figs 7 and 8) (Zeiss, Oberkochen, Germany). For flow cytometry, A549 cells in multiwell plates were detached using trypsin, washed in PBS and fixed in PBS 4% PFA. Cells

were then saturated/permabilized with PBS 5% FCS, 0.5% saponin, 0.05% Sodium Azide and stained with anti-Flag-APC conjugated mAb or anti-HA-FITC conjugated mAb for 30 min (Miltenyi). After PBS wash, cells were fixed with PBS 1% PFA and readily analyzed by flow cytometry using an LSR Fortessa cytometer (BD Biosciences). IFA and FACS data were respectively analyzed using ImageJ and FlowJo software (TriStar).

## Immunoprecipitation and Western Blot analysis

A549 or HEK293T cells were lysed in coIP buffer (25 mM TrisHCl pH7.4, 1% NP40, 150 mM NaCl, 1 mM EDTA, 5% Glycerol supplemented with cOmplete EDTA free protease inhibitor and PhosSTOP phosphatase inhibitor cocktails (Sigma)) for 30 min on ice. Cell lysates were centrifuged for 10 min, 10000 g at +4°C to pellet cellular debris and Whole Cell Extracts (WCE) were mixed with anti-Flag magnetic beads (M2 clone, Sigma) for immunoprecipitation (IP). Beads were then washed with coIP buffer and Flag tagged proteins were eluted with the addition of 1X-Flag peptide (Sigma). After SDS-PAGE separation and transfer on PVDF membranes, proteins from WCE and IP eluates were analyzed by Western Blot with the following antibodies: HRP conjugated anti-Flag (clone M2, Sigma), anti-HA (clone GG8-1F3.3.1, Miltenyi) and anti-β-actin (clone AC-15, Sigma); monoclonal antibody to GP2 (clone KL-AV-1B3, [69]); polyclonal rabbit anti-Z matrix protein (Agro-Bio, France), anti-NMT1 (GeneTex and Novus Biologicals), anti-NMT2 (Novus Biologicals), goat anti-HSP70 (sc1060, Santa Cruz) and mouse anti-LC3b (L7543, Sigma). Donkey anti goat-, goat anti mouse- and anti-rabbit–HRP conjugated were from Jackson Immuno Research. HRP substrates were SuperSignal West Dura (Pierce, Thermo Fisher) and Clarity Max Western ECL substrate (Bio-Rad). Images were captured with an ImageQuant Las4000 (General Electric) or a ChemiDoc (BioRad).

## Cu$^I$- catalyzed azide-alkyne cycloaddition ("click" reaction) and in gel fluorescence

The AzC12 myristic acid analogue diluted in DMSO was added to the cell culture medium of cells at a final concentration of 10 μM or 20 μM (C10268, Invitrogen). Alkyne-AF647 (Alk-AF647, 10 mM) CLK-1301A) and click chemistry reagents CuSO$_4$ (100 mM), Tris(3-hydroxy-propyltriazolylméthyl)amine (THPTA, 250 mM) and Na-Ascorbate (1 M) were prepared as recommended by the manufacturer (Jena Biosciences). To perform the biorthogonal ligation, cells were lysed in 1% NP40, 100 mM Na$_2$HPO$_4$ pH 7.0 buffer on ice for 30 min. After centrifugation 10 min at 10000 g at +4°C, WCE were collected and proteins were subjected to Flag immunoprecipitation (as described above). The "click reaction" was set in a 100 μL final volume with the addition of the following reagents: 94 μL of reaction buffer diluted WCE or IP eluates, 1 μL of Alk-AF647 (10μM final), a 3 μL mix of CuSO$_4$ (1 μL, 100 μM final) + THPTA (2 μL, 1 mM final) and 2 μL of Na-Ascorbate (5 mM final). After the addition of each reagent, samples were quickly vortexed and finally incubated for 1 h at 37°C in the dark and the reaction stopped with the addition 400 μL of acetone. After incubated 1h at -20°C, proteins were precipitated by centrifugation 30 min at 20000 g +4°C, supernatants removed and pellets air dried. Proteins were resuspended in PBS 2% SDS, 10 mM EDTA. After SDS-PAGE separation, the "in gel fluorescence" was detected with a DLx Odyssey and analyzed with Image Studio Lite (LI-COR).

## Protein tandem mass spectrometry

A549 cells were infected with rMOPV Z-Flag at MOI 1 for 1 h and treated or not with IMP1088 (50 nM). 48 h later, WCE were collected. Samples from three independent experiments were collected. For each sample, 25 μg of proteins were treated and proteolyzed with

trypsin as previously described [70]. The resulting peptides were analyzed in data-dependent acquisition mode with a Q-Exactive HF tandem mass spectrometer (Thermo) coupled to an Ultimate 3000 nanoRSLC nano LC system (Dionex-LC Packings), essentially as previously described [71]. They were resolved on an Acclaim PepMap100 C18 nano column (3 μm bead size, 100 Å pore size, 75 μm internal diameter, 15 cm length, Thermo) at a 0.2 μl/min flow rate with a 4–20% gradient of acetonitrile for 100 min followed by a 20%-32% gradient of acetonitrile for 20 min in presence of 0.1% formic acid. Full-scan mass spectra were recorded from $m/z$ 350 to 1800 at a resolution of 60,000 while MS/MS scans were acquired at a resolution of 15,000. MS/MS spectra were assigned to peptide sequences using the MASCOT search engine (version 2.5.1, Matrix Science) against a database corresponding to the Swissprot Human proteins supplemented with the Mopeia virus proteins. The following modifications were taken into account: carbamidomethyl of cysteines as fixed modification, oxidation of methionines, deamidation of asparagines and glutamines, and myristoylation of N-terminal glycine residues as possible modifications. Proteins were identified when at least two different peptide sequences were assigned (FDR below 1%). Comparative proteomics was performed as previously described [72] on the basis of the number of MS/MS spectra assigned per protein.

## Statistical analysis

Statistical analyses were performed using GraphPad Prism (GraphPad Software, La Jolla, CA, USA). Data were analyzed with One-Way ANOVA with a Bonferroni posttest. Statistical significance is defined as $p$ value <0.05 (*), <0.01 (**) and <0.001 (***).

## Supporting information

**S1 Fig. Characterization of the Z/NMT interaction.** A) The Z matrix protein of LASV and JUNV interact with NMT1 and NMT2. HEK293T cells were cotransfected for 24 h with the indicated plasmids before cell lysis and Flag immunoprecipitation. WCE and IP products were analyzed by Western Blot for the presence of Flag (Z matrix protein of LASV and JUNV), HA (NMT1, NMT2 and ITCH) and β-actin. B) Sequence alignment of the ARF6 N-terminal protein, a known K3 myristoylated human protein and the Z matrix N-terminal of MOPV, LASV (Josiah), LCMV (WE), LUJV, JUNV (Espindola), MACV (Carvallo) and GTOV (INH-95551). The G2 residue is highlighted in blue and the K3 residue in red. C) Evidence for the absence of K3 myristoylation of the MOPV Z matrix protein. HEK293T cells were transfected with control plasmids or plasmids expressing the WT, G2A or G2A/K3A mutants of the MOPV Z matrix protein and treated with AzC12 (10μM). WCE were submitted to Alk-AF647 bioorthogonal ligation or Western Blot for the detection of Flag (the Z matrix protein) and β-actin. Results in A and C) are representative of two independent experiments.
(TIF)

**S2 Fig. IMP1088 and DDD85646 inhibit late stages of *Mammarenavirus* biology during infection.** A-C) Curve fitted analyses of the results from Fig 4A–4C. Results for MOPV (A), LASV (B) and MACV (C) were obtained after normalization to the Mock condition and expressed as percentage of infectious titers to Mock. D) IC50 for IMP1088, DDD85646 and C75 for MOPV, LASV and MACV in A459 cells. E) A549 cells were treated or not with IMP1088 (20nM), DDD85646 or C75 (500nM) for 1 h before infection (Pre-infection), during infection with rMOPV WT at MOI 0.01 for 1 h (Infection only), after infection (for 48 h, Post-infection) or throughout the experiment (Complete). Cell culture supernatant were collected and titrated. The results of four independent experiments are expressed as mean +/- SEM to

the viral titer (FFU/mL).
(TIF)

**S3 Fig. IMP1088 treatment does not trigger inherent IFN response.** A549 cells were treated or not with IMP1088 (50nM) and infected or not with rMOPV Z-Flag at MOI 0.01 for 48 h. Total RNA and WCE were collected at 0 h, 24 h and 48 h post infection. mRNA levels for β-IFN were quantified and standardized to mRNA GAPDH levels. Results are mean +/- SEM of three independent experiments.
(TIF)

**S4 Fig. Importance of the G2A mutation in the GPC and Z ORFs of rMOPV.** (A) Results of the rescue and plaque phenotypes of WT, GPC G2A, Z G2A and GPC/Z G2A rMOPV. (B) Comparison of the kinetics' growth of WT and GPC G2A rMPOV in Vero E6 cells. Cells were infected at MOI 0.01 and supernatants were collected for 96 h and titrated. Results from technical triplicates are represented as mean +/- SEM to the viral titer (FFU/mL).
(TIF)

**S1 Data. xlsx file contains all numerical data of each figure of the study.**
(XLSX)

**S2 Data. PDF file contains all WB and IFA pictures of the study.**
(PDF)

## Acknowledgments

We thank Dr Blaise Lafoux for technical help with the LSM 980 confocal microscope, Dr Olivier Reynard for the goat ant-HSP70 antibody and Dr Kodie Noy for proof-reading of the manuscript. We are grateful to the BSL-4 team members for the help provided throughout this work.

## Author Contributions

**Conceptualization:** Xavier Carnec, Mathieu Mateo, Sylvain Baize.

**Data curation:** Xavier Carnec, Jean Armengaud.

**Formal analysis:** Xavier Carnec, Jean Armengaud.

**Funding acquisition:** Sylvain Baize.

**Investigation:** Xavier Carnec, Virginie Borges-Cardoso, Stéphanie Reynard, Jean-Charles Gaillard.

**Methodology:** Xavier Carnec, Mathieu Mateo.

**Project administration:** Xavier Carnec, Sylvain Baize.

**Resources:** Xavier Carnec, Heinrich Kowalski, Jean Armengaud.

**Supervision:** Sylvain Baize.

**Validation:** Xavier Carnec.

**Visualization:** Xavier Carnec.

**Writing – original draft:** Xavier Carnec.

**Writing – review & editing:** Xavier Carnec, Mathieu Mateo, Jean Armengaud, Sylvain Baize.

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
