## [Decision Letter · Decision Letter 0]

26 Aug 2024

Dear Dr. Baize,

Thank you very much for submitting your manuscript "targeting n-myristoyltransferases promotes a pan-mammarenavirus inhibition through the degradation of the Z matrix protein" for consideration at PLOS Pathogens. As with all papers reviewed by the journal, your manuscript was reviewed by members of the editorial board and by several independent reviewers. In light of the reviews (below this email), we would like to invite the resubmission of a significantly-revised version that takes into account the reviewers' comments.

Thank you for submitting your work for consideration of publication in PLoS Pathogens. Both reviewers think that "the experimental section of the paper has been superbly executed and the results have been clearly presented and nicely illustrated." That being said, there are some additional experiments and controls that still need to be included to improve the overall quality of the work. Please consider their comments, suggestions, and recommendations in both the Summary section and Major/Minor Issue sections carefully when revising the manuscript for a resubmission. Thank you.

We cannot make any decision about publication until we have seen the revised manuscript and your response to the reviewers' comments. Your revised manuscript is also likely to be sent to reviewers for further evaluation.

Sincerely,

Hinh Ly, Ph.D.

Guest Editor

PLOS Pathogens

Benhur Lee

Section Editor

PLOS Pathogens

Michael Malim

Editor-in-Chief

PLOS Pathogens

orcid.org/0000-0002-7699-2064

Thank you for submitting your work for consideration of publication in PLoS Pathogens. Both reviewers think that "the experimental section of the paper has been superbly executed and the results have been clearly presented and nicely illustrated." That being said, there are some additional experiments and controls that still need to be included to improve the overall quality of the work. Please consider their comments, suggestions, and recommendations in both the Summary section and Major/Minor Issue sections carefully when revising the manuscript for a resubmission. Thank you.

Reviewer's Responses to Questions

**Part I - Summary**

Reviewer #1: The study by Carnec et al demonstrates that arenaviral Z matrix proteins interact with cellular N-Myristoyltransferases 1 and 2 to become myristoylated. Knockout experiments suggest that while both NMT1 and NMT2 can modify the Z protein, the more abundant NMT1 was more biologicially important. This study builds on previous work that observed both Z and SSP myristoylation is important for arenaviral particle abundance and infectivity. Similar to the observations seen when genetically removing the Z myristoylation site, Carnec et al find two NMT inhibitors, IMP1088 and DDD85646, robustly inhibit particle production. They suggest the inhibitor leads to degradation of Z in a proteasome and autophagy independent manner, but this experiment lacks some controls to draw this conclusion. The study is well written and executed well, but a few controls are needed to fully support the conclusions. Overall the study provides additional evidence supporting acylation of arenavirus proteins are critical for their reproduction.

Reviewer #2: This work by Carnec and colleagues investigates the effect of pharmacological inhibition of host cell N-myristoyltransferases (NMT1 and NMT2) on mammarenavirus replication in cultured cells. Published evidence has shown that the mammarenavirus matrix Z protein and the stable signal peptide (SSP), generated via co- translational processing of the glycoprotein precursor GPC, undergo N-terminal myristoylation by host cell N-myristoyltransferases. Importantly, G2A mutations that prevent myristoylation of Z or SSP have been shown to affect Z mediated virus budding and GP2 mediated fusion activity required to complete the mammarenavirus cell entry process. Although the importance of Z and SSP myristoylation on mammarenavirus multiplication has been documented, these earlier studies were done using pharmacological agents that subsequent have been shown to exert their effects on a rather non-specific and off-target manner. Hence, the significance of the studies by Carnec and colleagues examining the role of N-myristoylation in the mammarenavirus life cycle using the validated on-target NMT1/2 inhibitors DD85646 and IMP1088.

The experimental section of the paper has been superbly executed and the results have been clearly presented and nicely illustrated. The authors should be commended for the extensive and high-quality work done, which supports the main message of the paper, namely that pharmacological inhibition of NMT1 and NMT2 using the validated on-target inhibitors, DD85646 and IMP1088, results in a potent dose-dependent inhibition of mammarenavirus multiplication in cultured cells that is associated with the matrix Z protein being targeted for degradation. These findings represent a significant contribution to our current understanding of mammarenavirus biology and provide also a strong support for investigating the use of NMT1/2 inhibitors as candidate host-directed antivirals to treat infections by human pathogenic mammarenaviruses.

An aspect that needs additional consideration is whether the known role of N-myristoylation in regulation of the host cell innate immunity could have influenced some to the reported findings. Does treatment with DDD85646 or IMP1088 affects the type 1 interferon response, which could, in turn, affect virus multiplication? Related to this issue, it would be informative to assess whether pharmacological inhibition of NMT1/2 can contribute to PARP and mitochondrial stress mediated antiviral response. It is unclear why the authors have not examined whether treatment with DDD85646 or IMP1088 also results in the expected lack of SSP myristoylation and the expected impact in GP-mediated cell fusion during the virus cell entry process.

**Part II – Major Issues: Key Experiments Required for Acceptance**

Reviewer #1: Line 209: Figure 3. Please explain how “These results seemed confirmed at the protein level in the matching WCE.” The immunoblots appear similar and based on the methods, different antibodies were used to detect NMT1 and NMT2. Without detailed antibody affinity information can you ever compare the overall expression levels of two different protein on a western? You can claim they are produced similarly across the time scale, but little more.

Why is HA-NMT2 in a separate blot in part B? It makes it difficult to assess relative expression levels.

In part C, could the “high molecular weight version of NMT2” simply be some low cross reactivity of NMT1 with the NMT2 antibody? How specific are the antibodies? Are the proteins the same size?

Figure 4: I was slightly confused about the inclusion of C75 in the assay. While it clearly did not inhibit the arenaviruses, it also did not alter yellow fever titers. Previous work has suggested that flaviviruses require FASN activity to efficiently replicate. Could C75 simply just not be active? Nothing to be too concerned about with your conclusions, just may want to ensure that C75 is actually inhibiting FASN to include it over a simple DMSO control.

Panel A has so many lines it is somewhat difficult to follow each, can you make the YFV line a different color or dashed lines to help?

Figure 5. Part A – the three gels are not aligned well. I know the bottom of the gels spread making it tricky, but maybe adding lane markers (1-14) on the top of each one would help. Part B has some missing labels on the right hand side (at least on my copy). I assume the top is an IP and bottom in WCE. Was part B also an anti-FLAP IP? Why does the FLAG IP not more closely match the one in A? Does the addition of the click chem reagents enhance the loss of Z produced or is the effect variable. The HA results are surprising with the amount of FLAG signal seen in the same lanes.

Figure 9. Is there a control you could probe for to demonstrate some other protein not getting myristoylated is rescued by MG132 (based on your rationale others have identified some). This would control for the drug treatment is working as expected. Currently you have a lack of a result, but no proof that things were working as expected to support your conclusion that it is not a proteasome degradation pathway.

Reviewer #2: Despite the excellent technical quality of the present work, there are some technical issues that need additional clarifications and discussion.

1) Levels of NMT1 and NMT2 interacting with Z appear to be rather low based on results shown in Fig. 1B. Moreover, steady state levels of NMT2 appear to be significantly lower than NMT1 levels, which does not seem to be affected by mammarenavirus infection. It is therefore unexpected that levels of NMT1 and NMT2 Co-IP with Z appear to be very similar. Is that possible that NMT2, despite being expressed at lower levels, has a stronger affinity for Z?

2) Western blot results shown in Fig 1D need quantification as levels of Z-FLAG protein appear to be quite different among samples.

3) Results shown in Fig. 5B: it is unclear why levels of Z-FLAG in mock-transfected cells appear to be similarly low to those in transfected cells. It should be also discussed why IMP1088 at 100 nM, a concentration that is effective inhibiting virus multiplication, seems to be unable to prevent the interaction of NMT1 or NMT2 with Z, required for inhibition of Z myristoylation.

4) Results shown in Fig. 4E would be strengthened by conducting a time of addition experiment using NH4Cl treatment to prevent the confounding associated with multiple rounds of infection.

5) Results shown in Fig. 7F seems to conflict with the role of Z as a negative regulator of the activity of the virus ribonucleoprotein complex that is responsible for directing the biosynthetic processes of replication and transcription of the viral genome.

6) Results presented in Fig. 9 do not provide unequivocal evidence supporting the conclusion that Z degradation is a proteasome-independent event. Results from the incorporation of an MG132 dose-response, and a control of a protein known to be targeted for degradation upon N-myristoylation inhibitor can help to clarify this issue.

7) Several of the experiments will benefit from the incorporation of the G2A mutant Z protein, which does not undergo N-myristoylation.

8) The authors should comment on published work defining the interactome of JUNV Z protein where a large number of Z-host cell interacting proteins, but not NMT1/2 , were identified.

**Part III – Minor Issues: Editorial and Data Presentation Modifications**

Reviewer #1: Line 29: Change “If” to “While”

Line 104: Change “or” to “and”

Figure 1 A – the FLAG IP immunoblot and the WCE in gel fluorescence could be aligned and cropped better. The Z band is touching the edge of the crop in the top and in the bottom, the gel is misaligned and the wt lane is half missing.

Figure 1 B – The Flag immunoblot appears overly saturate (burnt bands) is there a shorted exposure? Similarly, although not quite as bad, the NMT2 blot for WCE?

Figure S1 part C and Line 227, 231, 353, 698 remove the “e” at the end of actin

Line 205: Would be helpful to add citations here again

Line 345: The test say rMOPVmCherryHA but the figure says mCherryFLAG. Were these cells actually stained or just monitored for mCherry in flow?

Figure 7 B: May want to alter the Axes to “% of flag or HA” or “ % of tag +” rather than (Flag/HA) I was confused and thought you were showing a ratio of FLAG/HA positive

Line 530: Wasn’t this data from HAP1 cells, not A549?

Reviewer #2: The paper is in overall well written, but the introduction section would benefit from some additional editing to eliminate unclear sentences like “Mammarenavirus genus family have a worldwide...”, and to clarify that LASV, JUNV and other hemorrhagic fever causing mammarenaviruses do not cause epidemics per se, but rather they are endogenous to specific geographic locations.

The authors should comment on the discussion section on the small molecule NMT inhibitor PCLX-001, which has been shown to be safe and well tolerated in humans, supporting the interest of exploring the repurposing of NMT inhibitors to treat infections by human pathogenic mammarenaviruses.

I would suggest to incorporate a brief comment in the M&M section of the paper stating that experiments with LASV, LUJV, JUNV and MACV have been done in appropriate BSL4 containment facilities.

PLOS authors have the option to publish the peer review history of their article (what does this mean?). If published, this will include your full peer review and any attached files.

Reviewer #1: No

Reviewer #2: No
---

## [Decision Letter · Decision Letter 1]

31 Oct 2024

Dear Dr. Baize,

We are pleased to inform you that your manuscript 'targeting n-myristoyltransferases promotes a pan-mammarenavirus inhibition through the degradation of the Z matrix protein' has been provisionally accepted for publication in PLOS Pathogens.

Best regards,

Hinh Ly, Ph.D.

Guest Editor

PLOS Pathogens

Benhur Lee

Section Editor

PLOS Pathogens

Michael Malim

Editor-in-Chief

PLOS Pathogens

orcid.org/0000-0002-7699-2064

Thank you for revising the manuscript based on the reviewers' original comments and suggestions. If possible, please consider incorporating responses (in the form of clarifications) to those additional comments provided by reviewer #1 into the final version of the manuscript.

Reviewer Comments (if any, and for reference):

Reviewer's Responses to Questions

**Part I - Summary**

Reviewer #1: This resubmission explores the cellular NMTs required for mammarenaviral replication. The established a Z myristoylation assay using click chemistry and demonstrate NMT1 is the predominant NMT needed to modify Z. Further more they demonstrate NMT inhibitors block budding of a panel of arenaviruses by block Z myristolyation. NMT inhibitors led to the degradation of Z, but not through the proteosome or autophagy pathways. The paper is well written and was very responsive to reviews.

Reviewer #2: This is a revised version of a paper submitted by Carnec and colleagues investigating the effect of pharmacological inhibition of host cell N-myristoyltransferases (NMT1 and NMT2) on mammarenavirus replication in cultured cells. As I indicated in my review of the originally submitted paper, the findings presented in the paper represent a significant contribution to our current understanding of mammarenavirus biology and provide also a strong support for investigating the use of NMT1/2 inhibitors as candidate host-directed antivirals to treat infections by human pathogenic mammarenaviruses. The quality of the experimental work in the originally submitted paper was excellent and the results clearly presented and nicely illustrated.

In this revised version of their paper, the authors have adequately addressed the issues and questions I raised during the review of the originally submitted paper. However, there some minor issues that require some additional clarification. For the sake of the structure of the review format used by PPATHOGENES, I have listed them under Minor Issues.

**Part II – Major Issues: Key Experiments Required for Acceptance**

Reviewer #1: Previous concerns have been adequately addressed

Reviewer #2: In this revised version of their paper, the authors have adequately addressed all the major issues I raised during the review of the originally submitted paper. I do not have additional comments regarding the scientific content of this revised version of the paper.

**Part III – Minor Issues: Editorial and Data Presentation Modifications**

Reviewer #1: None - previous comments related to the figure labels etc have been addressed.

Reviewer #2: 1. The results presented in S3 Fig and the interpretation provided by the authors would be strengthened by incorporating images showing the numbers of infected and non-infected cells at the experimental endpoint.

2. I understand that the section under WCE in Fig 1D of the revised version of the paper correspond to a re-run of the same samples? Anything different in this re-run that account for the significant differences in the anti-flag immunoblots?

3. The results presented in Fig. 4E do not distinguish between the effects of NMT inhibitors on a cell entry or post-entry event of the virus life cycle. This is better addressed using either a single cycle infectious virus or an inhibitor of endosome acidification.

PLOS authors have the option to publish the peer review history of their article (what does this mean?). If published, this will include your full peer review and any attached files.

Reviewer #1: No

Reviewer #2: No

---

## [Editor Report · Acceptance letter]

26 Nov 2024

Dear Dr. Baize,

We are delighted to inform you that your manuscript, "Targeting n-myristoyltransferases promotes a pan-mammarenavirus inhibition through the degradation of the Z matrix protein," has been formally accepted for publication in PLOS Pathogens.

Best regards,

Michael Malim

Editor-in-Chief

PLOS Pathogens

orcid.org/0000-0002-7699-2064